# Individual environmental niches in mobile organisms

Ben S. Carlson [1,2 ✉], Shay Rotics[3,4], Ran Nathan[4], Martin Wikelski [5,6] & Walter Jetz [1,2]

Individual variation is increasingly recognized as a central component of ecological processes, but its role in structuring environmental niche associations remains largely unknown. Species' responses to environmental conditions are ultimately determined by the niches of single individuals, yet environmental associations are typically captured only at the level of species. Here, we develop scenarios for how individual variation may combine to define the compound environmental niche of populations, use extensive movement data to document individual environmental niche variation, test associated hypotheses of niche configuration, and examine the consistency of individual niches over time. For 45 individual white storks (*Ciconia ciconia*; 116 individual-year combinations), we uncover high variability in individual environmental associations, consistency of individual niches over time, and moderate to strong niche specialization. Within populations, environmental niches follow a nested pattern, with individuals arranged along a specialist-to-generalist gradient. These results reject common assumptions of individual niche equivalency among conspecifics, as well as the separation of individual niches into disparate parts of environmental space. These findings underscore the need for a more thorough consideration of individualistic environmental responses in global change research.

[1] Department of Ecology and Evolutionary Biology, Yale University, New Haven, CT, USA. [2] Center for Biodiversity and Global Change, Yale University, New Haven, CT, USA. [3] Department of Zoology, University of Cambridge, Cambridge, UK. [4] Movement Ecology Laboratory, Department of Ecology, Evolution and Behavior, Alexander Silberman Institute of Life Sciences, The Hebrew University of Jerusalem, Jerusalem, Israel. [5] Department of Migration, Max Planck Institute of Animal Behavior, Radolfzell, Germany. [6] Centre for the Advanced Study of Collective Behaviour, University of Konstanz, Konstanz, Germany. ✉email: ben.carlson@yale.edu

Climate and land-use change are impacting global biodiversity, making it a key scientific priority to predict how animal populations will be forced to move, adapt, or go extinct[1–5]. In assessing responses to global change, the importance of within-species variation is increasingly recognized because intraspecific variation mediates responses through a number of factors such as demographic growth rates[6,7], local adaptation[8], dispersal capabilities[9], range dynamics[10], competition and coexistence[11,12], eco-evolutionary dynamics[13], and ecosystem function[14].

An increasingly consequential form of within-species variation is niche specialization, which occurs when an individual's niche breadth is narrow relative to its population's niche breadth[15–17]. Following Van Valen's[18] seminal contribution, recent studies show that individual niche specialization is widespread in a diversity of taxa including insects[19], fishes[20], amphibians[21], reptiles[22], mammals[23], and birds[24]. The degree of specialization within a population influences the strength of competition and predator–prey interactions, demographic rates, and eco-evolutionary dynamics[15]. Thus, understanding the fate of populations under global change requires quantifying this important form of intraspecific variation.

Virtually all evidence for niche specialization is based on resource axes such as prey type or size[15,17], known as Eltonian niche axes[25], whereas global change and biogeographic studies usually estimate niches using climate or landcover, known as environmental or Grinnellian niche axes[25–27]. The concept of an individual niche is still in need of formal characterization. Here, we employ a working definition: an individual Grinnellian niche is the set of all points in environmental (niche) space, as defined by Grinnellian axes, that permit an individual to survive and reproduce. Typical approaches to putatively quantify Grinnellian niches, such as species distribution models (SDMs), ignore intraspecific variation and estimate a single species-wide response[26]. Thus, a central yet unresolved question is whether the documented niche specialization of animals in Eltonian niche space also extends to Grinnellian niches.

It is important to directly assess Grinnellian niches and not simply assume that individual niche patterns follow those of Eltonian niches. Although we expect that an individual's Grinnellian and Eltonian niches are interconnected, this relationship is complicated through a complex set of factors involving the individual's (and, in secondary consumers, putative prey's) behavioral response to the environment, and the scale at which environmental associations are assessed. For example, an individual that focuses on a single, generalist prey species will have a narrow Eltonian niche but a wide Grinnellian niche. Lack of a straightforward relationship between individual Grinnellian and Eltonian niches highlight the need to directly assess individual Grinnellian niches. At least two processes suggest conditions in which specialization in Grinnellian niches may occur. First, foraging theory predicts that individuals will tend to specialize on different prey types when they have different rank order for diet preferences, optimization criteria, or when experience leads to more efficient exploitation of specific prey items[17,28]. In generalist species, this can result in differential use of habitats if specific prey types are found in distinct habitats[29,30]. Second, animals might have differential focus on the habitats themselves, as opposed to specific prey, due to natal induction[31,32], because they have developed foraging techniques that are more efficient in specific habitats, or because individuals choose alternative habitats due to local competition[33]. However, these hypothesized drivers of Grinnellian niche specialization are context-, condition- and scale-dependent. For example, if prey species do not covary with environments at the scale at which the environments are measured, a signal of Eltonian specialization will not be reflected in the Grinnellian niche. Thus, direct assessment of individual Grinnellian niches is an important undertaking.

Popular approaches to investigate individual relationships to Eltonian and Grinnellian variables have strengths and weaknesses in their ability to address the multi-dimensionality of niche configurations and the resulting implications under global change. Studies of Eltonian variables usefully focus on usage distributions, but are often univariate[15] or bivariate[23] (but see ref. [34]), limiting the ability of their methods to address the multivariate nature of environmental use. In addition, these approaches rarely examine the configuration of individual niches. On the other hand, studies of Grinnellian variables embrace a multivariate approach, but do not directly estimate usage distributions. Instead, these studies usually describe individual behavioral responses to the environment[35–38]. Although they represent important contributions to our understanding of the impact of global change, for example the fitness consequences of migration strategies under variable environments[35], these studies do not examine the multidimensional usage distribution of Grinnellian variables, or the niche as hypervolume, as originally envisioned by Hutchinson[39,40]; and they thus are unable to assess individual use and specialization within environmental space. For example, resource selection analysis, a popular method that is used to infer animal-environment relationships, provides an index that measures a particular behavior—selection or avoidance of a resource, relative to its availability[41]. Among-individual variation in these behavioral indices can be examined to elucidate consistent, among individual differences in prey selection[36], habitat selection[42,43], or used to identify clusters of behavioral tactics and their drivers[44]. However, such resource selection approaches are unable to characterize the position and breadth of the multidimensional set of environmental conditions that permit individuals to survive and reproduce. In addition, as single points without properties such as size or shape, behavioral indices have limited ability to characterize an individual's position in niche space relative to other individuals. Directly representing individual Grinnellian niches as hypervolumes, based on usage distributions, would allow geometric interpretation of environmental relationships not possible using resource selection, including the size, shape, or relative location of an individual's niche. Furthermore, hypervolume representation allows direct examination of individual niches in environmental space, which greatly aids the interpretation of individual and population environmental requirements.

It is often assumed that individuals discretely partition niche space[15,17,44,45], but measures of specialization usually provide a population-level index and lack information on how niches are configured relative to each other, e.g.[46]. We use four example scenarios to show that individual niches can take a number of configurations within a population (Fig. 1, also see ref. [47]). In the simplest scenario, niches are all highly similar and thus overlapping (Fig. 1a). This lack of statistical differences in niches is usually assumed by most species- or population-level models including SDMs[26], food webs[48], coexistence models[49], and mutualistic networks[50]. Alternatively, individual niches might be clustered into a limited number of groups or modules (Fig. 1b). A set of niches exhibits high modularity if the niches are organized into multiple groups in which there is high niche overlap among individuals within a group but these individuals have low niche overlap with individuals outside the group[51]. This pattern can occur if exploitation of a limited number of alternate resources require specialized search and handling skills[29]. In another scenario, individual niches might be nested across a specialist-to-generalist gradient (Fig. 1c). This pattern can occur if access to resources is governed by a dominance hierarchy[52], or if individuals have similar rank preferences but differ in the degree to

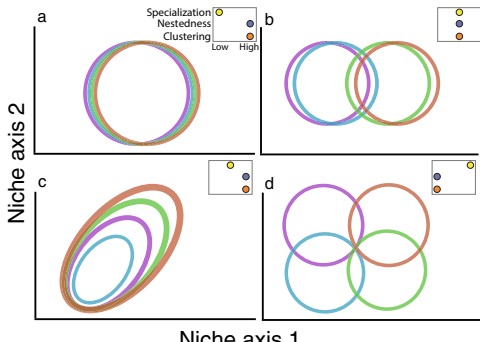

**Fig. 1 Example configurations of niches (circles/ellipses) of a population of four individuals along two resource axes.** Colors indicate different individuals. Niches might be all similar and overlapping (**a**), clustered into a limited number of groups (**b**), nested according to a generalist/specialist gradient (**c**), or are different and occupying separate parts of environment space (**d**). Each of the four scenarios implies different patterns of specialization, nestedness, and clustering—the three metrics used in this study to delineate realized configurations. See text for further details and interpretation.

which they will accept less-preferred resources[28,47]. Finally, individuals might all have distinct niches and occupy separate parts of environmental space (Fig. 1d). This pattern is commonly assumed in studies of individual variation in Eltonian niches[15,17], but it is unknown what pattern applies to Grinnellian niches.

These example individual niche configurations carry different implications for population ecology and conservation under environmental change. For example, populations that are composed of limited groups of niches may indicate responses to important alternative resources that should be targeted for conservation (Fig. 1b). Or, depending on the nature of environmental change, populations composed of individuals with narrow, disparate niches (Fig. 1d) may be either more or less susceptible than populations that contain a specialist-to-generalist gradient of niches (Fig. 1c) or undifferentiated niches (Fig. 1a). Thus, to conserve populations under global change, it is important to have a deeper understanding of niche configuration than simply knowing whether a population is composed of generalists or not.

Scrutiny and identification of intraspecific niche configurations at landscape and regional scales have been limited by suitable data and methodologies. Diet-based estimates of Eltonian specialization usually rely on painstaking field work[53,54], imposing limits on geographic and temporal extent. Advances in animal tracking technology, the availability of remotely sensed habitat information, and low-cost cloud computing now offer a new and more scalable means to quantify individual specialization. Although there have been important advances in using these types of data to estimate individual variation in behavior, including resource selection and repeatability, no study has represented individual niches in a multidimensional framework that allows examination of the geometric configuration among individual niches.

Here, we capitalize on opportunities to capture and compare environmental hypervolumes (niches) of 45 individual white storks (*Ciconia ciconia*) from three breeding populations in Germany over four years (2013–2016). Linking these data to high-resolution remote sensing data and using multivariate niche metrics, we ask: (1) Are individual Grinnellian niches more specialized than expected by chance? (2) Which hypothesized niche configuration (Fig. 1) is supported by the data, e.g. are Grinnellian niches similar, clustered, nested, or differentiated? And (3) what is the consistency of individual niches and population niche configurations across time?

## Results

We collected nearly one million GPS tracking-based breeding season locations from 45 individual White storks (*Ciconia ciconia*) in three populations over four years. We linked these locations to carefully selected, remotely sensed environmental variables that represent known habitat associations for the species. These variables, all captured at 30 m resolution, address key aspects of foraging habitat structure, quality, and access (16-day NDVI, percent of tree and bare ground cover, distance to built-up areas and forest; Figs. S6, S8–10, Table S1). In order to investigate the relevance of these variables and their suitability for a general assessment of individual environmental niches, we performed resource selection analysis using standardized (z-transformed) variables, and included two additional terms to adjust for distance to the nest and for a hypothesized interaction between NDVI and tree cover. Specifically, we required a statistically significant use (relative to availability) of an environmental variable as evidence that the variable represents a relevant niche axis. We found that a majority of individuals had significant associations, both preferential selection and avoidance, with all measured environmental conditions (Fig. S1). For example, in the Loburg population in 2015 ($n = 9$; Fig. S1), individuals selected foraging areas with lower vegetation greenness than the surrounding landscape while avoiding bare and forested areas, confirming this species' association with open grasslands, meadows, and pastures with short vegetation. However, both the strength and direction of selection for specific variables varied strongly among individuals. For example, some individuals tended to forage away from urban areas, others preferred proximity to these areas, and yet others demonstrated no selection. This pattern of statistically significant selection and strong among-individual variation in the strength and direction of selection is confirmed in two other populations and four other years ($n = 107$ individual-year combinations, Fig. S1).

We then used the five main environmental variables (and not the two additional variables, distance to the nest and NDVI percent tree cover interaction) to construct and compare individual hypervolumes. For visualization purposes, we used a dimension reduction technique (multidimensional scaling) to provide an initial, two-dimensional illustration of individual differentiation in niche space (Fig. 2b). Visually, some individuals occupy a much broader range of environments compared to other, more specialized individuals. In all populations and years, specialist and generalist individuals overlap in a core region of niche space. The more generalist individuals extend beyond this core environmental space to different magnitudes and into different and sometimes unique regions of multivariate niche space. This visual inspection in reduced two-dimensional space provides preliminarily support for the hypothesis that the storks' niches have a nested pattern (Fig. 1c).

Formal analysis using specialization, nestedness, and clustering metrics in five-dimensional niche space confirms this result. Specialization measures the average difference between the individual and population niche. Nestedness measures, for each pair of individuals, how much of the smaller niche is contained within the larger niche. Clustering is a weighted global clustering index based on network theory and measures the amount of modularity among individual niches. Each population had an overall high, if variable level of individual specialization (Fig. 2b). Similarly, individuals in all populations exhibit high levels of nestedness, confirming the visual appearance in the two-dimensional plot of smaller niches nested inside larger niches (Fig. 2c). Finally, a high clustering index indicates that pairwise overlap was generally high within population years and therefore lacked modularity, despite comparatively lower overlap among individuals' home ranges (Fig. 2a). Our clustering index rarely deviated from 1. To obtain a

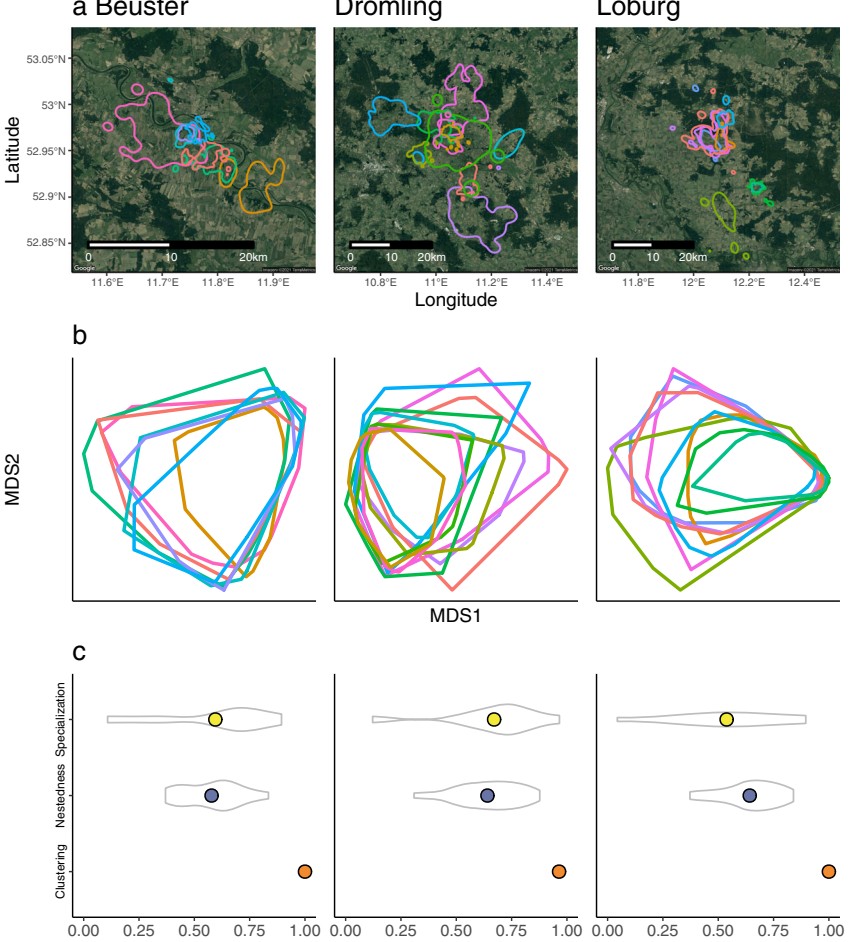

**Fig. 2 Home range and environmental niche configurations of 26 white storks (Ciconia ciconia) in three populations during the 2015 breeding season.**
**a** Individual home ranges, with lines representing 95% contours and colors identifying single individuals. **b**, **c** Geometric configuration and associated metrics of individual niches of the individuals shown in (**a**). Individuals are arranged on a specialist-to- generalist gradient, with individual niches extending into unique environmental space as their niche volume increases. See Fig. S2 for other years (2013–2016) which display similar patterns. **b** Visual representation of niche geometry and configuration in reduced two-dimensional space. Dimension reduction was performed on the estimated hypervolumes using non-metric multidimensional scaling (NMDS) on a subset of 1000 points from each hypervolume. Polygons represent the 95% outer contour of each individual niche. Colors and individuals are consistent with those in (**a**). **c** Specialization, nestedness, and clustering metrics for the three populations, calculated on the five-dimensional niches. All metrics range from 0 (low) to 1 (high). Colored points represent the population-level metric and violin plots characterize the distribution of individual or pairwise metrics for specialization and nestedness, respectively. See Fig. S3 for distributions of these metrics under three null models (further described in the methods section). Basemap images in (**a**): Google, ©2021 TerraMetrics.

value of clustering < 1, at least two (or more) niches need to have Jaccard overlap < 0.05. It is hypothetically possible that individual animals in our study partitioned environmental space to such a degree that there is very little overlap in their niches, but we don't find that this is true with our populations. Thus, we do not find any evidence for modularity in our study. Comparison against null models confirms that these results are driven by individual identity and individual resource selection preferences (Fig. S3, Data S1). In combination, these metrics signify individual niches clustered around a core set of environments, differentiated along a specialist-to-generalist gradient (Fig. 1c). They reject a pattern of individual specialization into disparate parts of environment space, as assumed by most studies of Eltonian niche specialization (Fig. 1d).

After observing similar resource selection and niche configuration patterns among all populations and years (Fig. S2), we conducted a formal analysis of the consistency of these patterns over time using the Intraclass Correlation Coefficient[55]. The ICC is a common metric of repeatability and measures the proportion of population variance explained by variance among individuals.

Individual resource selection and specialization had highly consistent, among-individual differences across years, as evidenced by high repeatability in selection coefficients for all seven environmental conditions estimated by resource selection analysis (mean: 0.39, range: 0.29–0.58), as well as for individual niche specialization (0.49, 95% CI [0.26, 0.65]). The observed repeatability for specialization rejects the null hypothesis when calculating repeatability under each of the three null models ($p < 0.01$; Fig. S7). Visual inspection confirms the stability of population-level specialization, nestedness, and clustering metrics among years (Fig. 3).

## Discussion

Individuals in our three study populations have specialized environmental niches. This confirms that a pattern commonly found in Eltonian niches[15,17] also applies to Grinnellian niches. However, in contrast to the frequent assumption of niche partitioning in diet space[15,17,45] (but see ref. [47]), we found that individual environmental niches are not configured into disparate

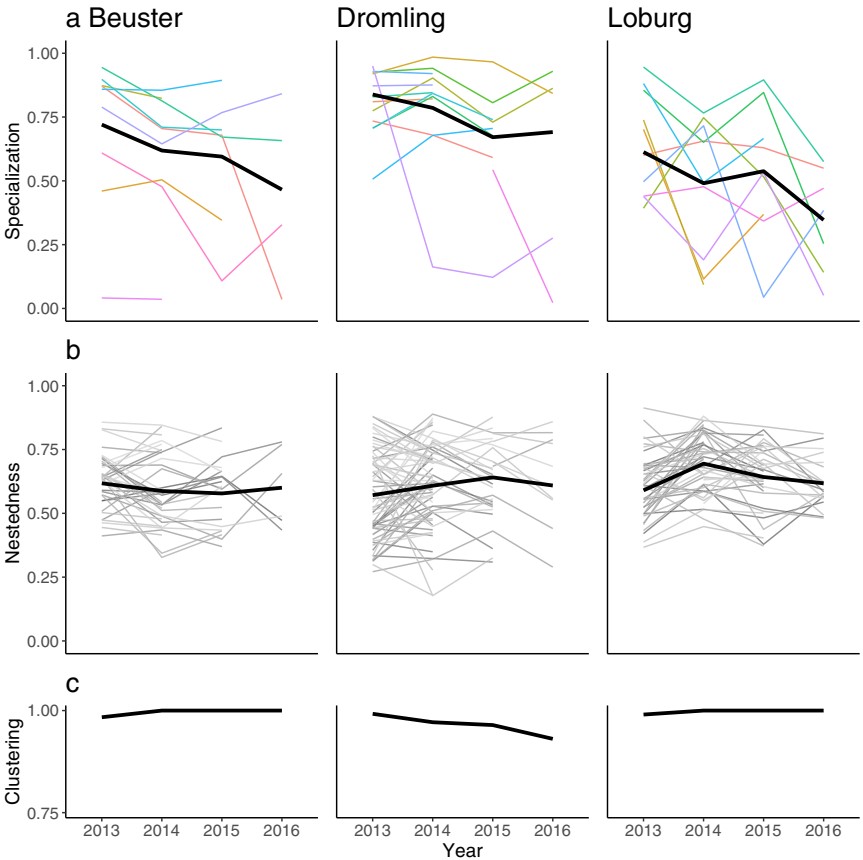

**Fig. 3 Individual and population niche configuration metrics over time.** Specialization (**a**), nestedness (**b**), and clustering (**c**) indices for three populations of white storks during the breeding season (Apr–Aug) over four years (2013–2016). Bold black lines represent population-level means. Thin colored/grey lines indicate individual (**a**) or pairwise (**b**) metrics. Individual colors in (**a**) follow Fig. 2. The specialization metric has a high overall repeatability ($R = 0.49$). All metrics range from 0 (low) to 1 (high).

regions of niche space. Instead, visual ordination and quantitative multidimensional metrics show that the configuration of individual environmental niches has a nested pattern (Figs. 1c, 2). In this configuration, individuals have a gradient of niche sizes. This size gradient is arranged such that all niches overlap a core environmental region, and larger niches tend to extend into unique parts of environmental space. Furthermore, environmental selection behavior and the level of individual specialization were consistent over the four-year study period (Fig. 3), indicating stability in individual niches and population-level configuration.

Individual variation in resource selection (Fig. S1) supports the hypothesis that behavior, specifically habitat selection, is an important proximate cause of specialization. Thus, the strength and direction of resource selection should have a relationship to the size, shape, and configuration of niches. For example, strong selection for an environmental variable is indicative of specialist behavior[56] and should result in narrower niche breadth. Alternatively, weaker selection, which is indicative of generalist behavior[56], means an animal is more likely to accept environments in proportion to their availability, thus resulting in relatively broad niche widths. We suggest, however, that the multivariate nature of niches can introduce subtle effects that require cautious interpretation which can be aided by jointly performing resource selection and hypervolume analysis. For example, if strong selection for one variable coincides with weak selection for another, this can result in the appearance of generalist behavior and a wide niche breadth for the second variable, when in fact a wide niche breadth is simply the result of non-

significant selection. Thus, it is important to simultaneously consider the behavioral aspects of resource selection and the geometry of hypervolumes when interpreting resource requirements.

Environmental niches in our study exhibited a nested pattern. Examples of nested patterns in dietary niches exist, although these are usually based on binary, bi-partite diet networks and not on hypervolume analysis[47,51,52,57–59]. Nested diets are often attributed to a "shared preferences" model, based on optimal foraging theory, in which competition forces individuals to seek less-preferred food items and individuals differ in their willingness to accept these items[28,57]. This nested-pattern model could extend to environmental niches, if broader diet breadth is associated with broader use of environmental conditions. A key component of foraging theory models is the contrast between the intensity of competitive pressure[51]. However, in our study, the level of nestedness was stable over time. This stability could be an indication that competitive pressure was also stable over time, or suggest an entirely different mechanism structuring niches.

Individuals showed consistent among-individual differences in resource selection and niche specialization over the four-year study period, implying some process may be structuring niche specialization and configuration. In some systems, social dominance hierarchy or the existence of distinct morphotypes causes consistent niche structure[15]. Breeding white storks often forage alone but are also known to form aggregations[60]. Although conditions exist for social dominance to occur while aggregating[61], we are unaware of any evidence for social dominance in white stork foraging. Furthermore, dominance effects, if

they occur, should affect the use of resources within a patch (e.g. feeding rates on an animal carcass[62,63]), whereas variation in environmental niches is due to differential use among patches. Thus, it is unlikely that dominance hierarchies play a significant role in environmental niche specialization. Likewise, we are unaware of the existence of distinct morphotypes (e.g., sympatric morphotypes in populations of lake trout[64]), thus it is unlikely that morphology is an important factor. Finally, in our populations, specialization is not driven by home range size (Fig. S4) or sex (Fig. S5). Instead, there are at least two plausible explanations for the consistent among individual differences in niches we find here. First, innate personality traits such as boldness or propensity for exploration might cause some individuals to use a wider range of environments than others[65]. Second, slowly developing or canalized foraging preferences such as natal induction or experience and skill in handling different prey types might influence the ability and willingness to exploit different environments. Finally, among-year consistency may in part be explained by the consistent selection of nest location across years. Our analysis of the differences in the environment available to each individual (Fig. S3) shows that a portion of niche specialization can be attributed to differences in environmental availability, e.g. due to selection of the nest site (level two selection, in Johnson's[66] four orders of selection framework). However, in most cases the hypothesis that specialization and nestedness are completely due to differences in available environments is rejected, indicating that even within available environments (i.e. level three selection), individual white storks display differential environmental use. Although we here are not able to partition the niche metrics between level two and level three selection, we underscore this as an important future research topic given the increased interest in scaling niches from individuals to populations and species.

Both the degree of specialization and the configuration of environmental niches we identify, if present more broadly in additional populations and taxa, have important implications for population responses to impending global change. We suggest that the degree of environmental niche specialization, in particular, has direct implications on the management and conservation of populations. First, populations are often managed by estimating an average response, but non-linear response due to specialization implies that the response of the population mean is not the same as the mean of the individual responses (i.e. Jenson's inequality[67]). In addition, SDMs, a prevalent tool in global change research, by design do not account for individual-level variation. Based on our findings we suggest this limitation may contribute to SDMs poor performance in transferring to different times or places[68,69], although we recognize that similar research will need to be conducted on additional populations and taxa to understand whether this is a widespread issue.

A second major implication for global change inference is given by the particular geometric configuration of niches (Fig. 1). Of the potential configurations, populations composed of generalist individuals (Fig. 1a) are likely the most resilient, because each individual uses the full range of environmental space. Populations with a highly modular configuration (Fig. 1b) may be less resilient because global change impacts are often directional and permanent (e.g. warmer climate, landcover conversion), thus impacting the environments underlying one or more of the modules. Populations composed of specialists (Fig. 1d) may have reduced resilience to global change, depending on the underlying mechanism causing specialization. If individuals have high behavioral plasticity, for example through a variety of innate behaviors or cognition, populations may react quickly. However, if specialization is due to traits more strongly linked to cultural or phenotypic evolution, populations will have a reduced capacity to

rapidly adapt to altered environments. Finally, populations composed of a range of specialist-to-generalist individuals (Fig. 1c) uniquely feature a core region of environmental space, and thus may be most impacted by changes to this core environment. Thus, identifying and conserving or augmenting this core environment might have outsized effects on population persistence.

Here, we have shown that three populations of white storks have consistently nested environmental niches. However, it is unknown whether this is a general pattern for other taxa. Individual variation in diet exists in a wide range of taxa[15,17], among animals with many different traits, so specialization and nestedness in environmental niches may also be common in nature. Future research should extend the analyses described here to additional taxa, especially to species with traits predicted to influence the degree of specialization. In addition, future studies can use this framework to investigate the causes and consequences of Grinnellian niche specialization and a nested configuration. The ability to extend these analyses to other taxa is facilitated by rapid growth in the data types we use in this study. Movement data are growing dramatically as tracking devices become smaller and less expensive, through initiatives such as Icarus[70] and the ATLAS tracking system[71], and are increasingly available through repositories such as Movebank[72] that facilitate the sharing and distribution of movement data. Remote sensing of the environment is also rapidly increasing in sensor capabilities, number of satellites, and data availability. Ongoing and upcoming Sentinel and Landsat satellite missions, as well as a proliferation of CubeSats can increasingly elucidate individual environmental niches by measuring environments ever closer to the grains that inform movement decisions[73–75]. Together, we expect these advances to improve our ability to quantify and understand individual niche structure and offer more mechanistic, individual-based prediction of the fates of populations under global change.

## Methods

**Study species and locations.** We estimated the degree of specialization in Grinnellian niches within three populations of white storks (See ref. [76] for additional information). The white stork is an excellent species to investigate specialization because the species has characteristics that should result in populations composed of specialized individuals[77]. White storks are generalist predators that forage in many different habitat types. White storks do not have age or size structure in access to resources, and are not strictly territorial but often have overlapping home ranges. They will at times forage together in the same patch but can also display local aggression, chasing conspecifics away from their foraging area or nest.

The three white stork populations are located in Northern Germany at sites near Beuster (52.94°N, 11.79°E), Drömling (52.49°N, 11.02°E), and Loburg (52.12° N, 12.09°E) (Fig. 2a). Each site features different landscape characteristics and has white stork populations with demonstrated individual variability in foraging choices[78]. The landcover around Beuster is agricultural but is dominated by a large river with associated riparian habitat. Drömling is located near a conservation area with a high proportion of marshland. The site at Loburg has higher urban density, and is primarily agricultural fields with one small stream.

**Tracking data.** We tagged 45 adult white storks with GPS and accelerometer sensors in order to measure fine-grained space use and behavior. The sensors recorded location and body acceleration every 5 min. GPS locations had a 50th percentile spatial accuracy of <3.6 m (50% of the points are within 3.6 m of the true location), and 95th percentile accuracy of <19 m. We tagged individuals starting in 2012 and data for some individuals continues to 2019, but for sufficient representation here we limit the analysis to 2013–2016. No bird was injured during the trapping and tagging procedure and all birds were released back to the wild. The research was carried out with the following permits from the authorized bodies to approve research in our study areas (i) the National Administrative Office of Sachsen-Anhalt, Germany, Division of Nature Conservation, 407.3.3/255.13-2248/ 2, (ii) the State Office for Environment, Health and Consumer Protection of Brandenburg, Germany, V3-2347-8-2012. In order to focus on the period in which individuals are sympatric, we manually segmented tracks into breeding and non-breeding periods by inspecting maps of daily movement and plots of net squared displacement[79]. We then further limited observations to April 1 through Aug 31, in

order to ensure that all individuals in a population had exposure to the same environments. We assigned discrete behavioral modes to each GPS location by relating field observations of behavior to accelerometer data using supervised classification. Thus each location is assigned to one of five behavioral classes: active flight, passive flight, sitting, standing/preening, and walking/pecking[76]. We defined observations classified as walking/pecking as foraging locations. We removed all locations not classified as foraging locations, as well as all locations within 100 m of the nest. In total 946,894 fixes informed the analyses, representing 116 individual years of data. All data are available in the Movebank data repository.

**Home range estimation**. To ensure that animals within each population had access to a similar habitat, we estimated home ranges using auto-correlated KDE[80] using the R package ctmm[81]. We visually inspected 95% contours and removed individuals in a particular year if they did not overlap with any other individual in that year.

**Niche assessment—overview**. We assessed individual niche specialization and configuration using a three-step process. First, we identified environmental covariates that we hypothesized would drive the selection of foraging locations and would therefore serve as individual niche axes. Second, we evaluated these axes by performing a form of resource selection analysis known as a step-selection function (SSF)[82]. These models reveal individual selection of environmental conditions relative to availability, and provide an assessment of the suitability of the environmental conditions for use as niche axes. Third, employing these identified axes, we used a multivariate kernel density approach[83] to estimate geometric niche objects in *n*-dimensional space. Unlike many other approaches that purport to estimate niches, this kernel density approach employs usage distributions to directly represent niches as hypervolume objects, which can be subjected to geometric set operations such as intersection or union. Finally, we used these hypervolumes to perform specialization and configuration analysis by using their geometric relationships to calculate metrics of specialization, nestedness, and clustering.

**Environmental conditions**. We selected five variables that capture the expert knowledge-based foraging niche of breeding white storks and also represent axes on which individuals likely specialize[84] (Table S1, Fig. S6, S8–10). Individual white storks vary in how they exploit anthropogenic resources[29,85] (e.g. proximity to urban landcover), open or woodland environments[86] (e.g. the amount of tree cover in a patch), and bare patches of ground[30] (e.g. the proportion of bare ground in a patch; often as earth worms). In addition, many individual animals vary in their preferences for edge habitats[87] (e.g. the proximity to forest), and in vegetative cover, height, and biomass[88] (e.g. as indexed by average NDVI in a patch). All variables are at 30 m spatial grain, which is small enough to capture both the high degree of heterogeneity in the landscape and the grain at which storks make foraging decisions. We standardized (i.e. z-transformed) all variables to have mean 0 and standard deviation 1. This results in units of standard deviations for all variables. This is the recommended procedure for linear models[89] as well for hypervolume estimation[83]. We standardized over the full dataset, in order to maintain comparability among individuals and sites. We used Google Earth Engine[90] to derive variables and to spatiotemporally associate environmental variables to foraging locations.

**Resource selection**. In order to provide more rigorous evidence about whether the environmental conditions we selected are niche axes, we performed resource selection analysis[91]. Resource selection analysis does not directly model niches, but instead we use it as tool to help evaluate niche axes. Thus, our use of resource selection analysis is a methodological step to empirically assess whether the environmental variables we chose are important to the individuals and thus can be considered niche axes. We interpreted statistically significant selection for or against a given environmental variable as evidence that the variable is, or is strongly correlated with, a niche axes for that individual. Resource selection analysis seeks to understand when animals use a particular combination of environments with more or less frequency than are available. Based on weighted distribution theory, a resource selection function $w(x)$ relates available resources $f_a(x)$ to used resources $f_u(x)$ through the relationship $f_u(x) \sim w(x) f_a(x)$[92]. The statistical model seeks to estimate the parameters of $w(x)$, which are covariates based on observed environmental variables. Traditional resource selection models assume that all locations within the modeling domain are equally accessible; a step-selection framework relaxes this assumption and instead dynamically updates the available background based on the characteristics of the movement path[82]. We modeled resource selection using a step-selection framework and conditional logistic regression models for each individual and year using the R package amt[93]. We used the five habitat variables described above. We included distance to nest as a covariate to adjust for differential habitat use that may occur due to nest proximity[94]. Additionally, we hypothesized that storks are attracted to patches of grassland with relatively high NDVI, but that this would not similarly be the case for forested patches with high NDVI. Thus, we should see a weaker response when a pixel with moderate to high NDVI also contains high percent tree cover. To account for this effect, we included a term that adjusts for interaction between NDVI and percent of

tree cover. The addition of these two additional terms (distance to nest and NDVI/ tree cover interaction) resulted in a total of seven terms in the SSF model, although we only use the five main terms as niche axes in our estimation of hypervolumes.

**Niche estimation**. We estimated individual foraging niches during the breeding season as hypervolumes. We used the environmental variables identified through step-selection analysis as hypervolume axes. We included conditions as niche axes as long as more than half of the individuals in a population had statistically significant selection (confidence intervals did not overlap 0) for most of the years. After using step selection to identify niche axes, we did not use step selection in our downstream analysis but instead focused on representing niches as hypervolumes. We directly used the standardized environmental data to estimate hypervolumes, using a kernel density approach that estimates the geometric shape of the niche in n-dimensional space[83]. In order to have a uniform sample size for all individuals, we randomly subsampled observations to 2000 foraging locations from each individual for each breeding season. This number of observations is well above the recommended minimum number of ~150, based on observations > exp(axes), where we use five axes[83]. These geometric niche objects can be subjected to set operations such as volume, union, and intersection, which we used to calculate specialization and niche configuration metrics.

**Specialization and niche configuration**. To measure the degree of specialization for each individual, we follow others[15,17,23,95] and use niche volume as a measure of niche breadth (for discussion of alternative definitions, see ref. [96]). Specifically, we calculated the specialization index as one minus individual niche breadth divided by population niche breadth[23]. That is

$$1 - \frac{\mathrm{Vol}(A)}{\mathrm{Vol}(\mathrm{Pop}_A)} \quad (1)$$

where $\mathrm{Vol}(A)$ is the volume of the hypervolume for individual A and $\mathrm{Vol}(\mathrm{Pop}_A)$ is the volume of the population hypervolume to which individual A belongs. We estimated population niches as the union of all individual niches, for each population and year. The specialization index ranges from 0 (least specialized) to 1 (most specialized). We used the mean of all individual specialization scores as a population-level index of specialization[46].

In order to further quantify niche configuration, we adapted two additional metrics: a nestedness index and a clustering index[51,97]. The nestedness index is performed pairwise for individuals within each population/year and measures the amount of the smaller niche that is nested within the larger niche. Originally proposed for diet matrices[98], we adapted it for use with hypervolumes as

$$\frac{\mathrm{Vol}(A \cap B)}{\min(\mathrm{Vol}(A), \mathrm{Vol}(B))} \quad (2)$$

where A and B are pairs of hypervolumes and $\mathrm{Vol}(X)$ is the volume of hypervolume X.

Likewise, we adapted a weighted global clustering index[97] for use with hypervolumes. In the context of niche configuration, this metric provides information about the degree of modularity. We constructed networks in which the nodes represented individuals and edges between nodes were weighted by the degree of niche overlap. We measured niche overlap using the Jaccard overlap metric, $\mathrm{Vol}(A \cap B)/\mathrm{Vol}(A \cup B)$, where A and B are niches of two individuals. If two individuals had niche overlap <5%, we considered these nodes unconnected and set the weight to 0. We used the R package tnet[99] to calculate the clustering index for each population and year.

**Null models**. We employed three null models to understand the role that variation in individual identity and available environments played in the observed level of niche specialization and configuration.

First, to understand the role that individual identity played in observed patterns of niche configuration, we randomly drew samples, with replacement, from the set of all observations in each population/year. We then constructed hypervolumes from these randomized niches and calculated niche metrics as described above. We repeated this process one hundred times to understand the distribution of specialization, nestedness, and clustering present when individual identity is not considered.

Second, to understand the role that differences in available environment played, we sampled the environment available to each individual. Storks have immense movement and navigation capacity, but breeding individuals are constrained in the distance they can travel from the nest by their need to feed their chicks and defend the nest site from predators and other storks. These constraints change over the breeding period. As chicks mature, parents can travel farther from the nest. Our goal was that the available environment should reflect these constraints, which are not captured by our environment variables, but should not include any of the factors that are captured by our environmental variables. Therefore, for each individual, we calculated the 95-percentile distance from the nest for each two-week period, drew a buffer around the individual's nest at this distance, and randomly sampled within each buffer. The environmental conditions of all random points sampled from the buffers of each individual represents the environment available to that individual during the breeding period. We used a bootstrap approach to randomly sample, with replacement, within each individual's available

environment, constructed hypervolumes and then calculated specialization, nestedness, and clustering metrics. We repeated this procedure one hundred times.

Finally, we combined aspects of the first two null models in a third. Specifically, we used a population-level resource selection function, as in the first null model, but constrained the available distribution using the buffers described in the second null model. We implemented the third null model by interpreting resource selection according to weighted distribution theory (see section Resource Selection, above). In a typical resource selection analysis, we use $f_u$ and $f_a$ to estimate $w$. For the null model, we take as $w$ the back-transformed mean of the selection coefficients for the individuals in each population/year. We then use this to sample from $f_a$ in order to generate $f_u$. By sampling from the available backgrounds at each nest site according to a population-level RSF, we produce usage distributions for each site as though they are generated by the same individual (in this case, the average stork). We then used these distributions to construct hypervolumes as described above.

**Repeatability**. We performed repeatability analysis to determine if consistent, among-individual differences exist in environmental selection and level of specialization. We computed repeatability as the Intraclass Correlation Coefficient[55] as the proportion of total variance explained by among-individual variance:

$$R = \frac{V_A}{V_A + V_E} \qquad (3)$$

where $V_A$ represents among-individual variance and $V_E$ represents within-individual variance.

We used the package rptR[55] to calculate repeatability and associated confidence intervals. Although repeatability analysis is often used to examine consistent individual differences in traits linked to personality (e.g. boldness, aggression) repeatability analysis is not limited to these traits and has been used to assess consistent among-individual differences in habitat selection[42,43] and movement metrics[100]. Here, we use repeatability analysis to assess habitat selection[42,43], but also uniquely assess consistent individual differences in niche specialization, which is enabled by our use of hypervolumes.

## Data availability
The movement data[101] generated in this study have been deposited in the Movebank Data Repository under accession code https://doi.org/10.5441/001.1.rj21g1p1.

## Code availability
The code[102] used in the analyses is available at the following public repository: https://doi.org/10.5281/zenodo.5032460.

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

## Acknowledgements

This research was funded by National Aeronautics and Space Administration (NASA) grant 80NSSC18K1404 to W.J. and B.C. and by the Max Planck Institute of Animal Behavior. We acknowledge funding of DIP grants (DFG) NA 846/1-1 and WI 3576/1-1

to R.N. and M.W. yielding the stork movement data. R.N. also acknowledges support from the Minerva Center for Movement Ecology and the Adelina and Massimo Della Pergola Chair of Life Sciences. We also acknowledge funding by the Deutsche Forschungsgemeinschaft (DFG, German Research Foundation) under Germany's Excellence Strategy—EXC 2117—422037984. We thank the Max Planck-Yale Center for Biodiversity Movement and Global Change for additional support.

## Author contributions

B.C. and W.J. conceived the study, designed the analyses and figures, and wrote the manuscript. B.C. conducted the analyses. S.R. and M.W. provided feedback on the study system, design and analyses. S.R., R.N., and M.W. provided data and commented on the manuscript.

## Competing interests

The authors declare no competing interests.
