## [Peer Review File · Nature Communications]

Reviewers' comments:

Reviewer #1 (Remarks to the Author):

The manuscript entitled „Individual environmental niches in mobile organisms“ by Carlson et al. uses movement data of 45 white storks to expand the concept of intra-specific variation in dietary niche breadth and composition (Eltonian niche) to intra-specific variation in environmental niche breadth and composition (Grinnellian niche). Traditionally, Grinnellian niches are typically estimated on the species level using species distribution models (SDM's). The authors here use resource selection functions (more specifically, step selection functions SSF) fit on individuals to show that storks differ in habitat selection with some storks using a wider range of habitats than others. The authors conclude that intraspecific variation in environmental niches exist and that white storks can be organized along a specialist to generalist gradient with all storks using a core of the same environment but some additionally also use other environments.

I generally agree with the premise of the paper and congratulate the authors for their elegant conceptual approach of combining the literature on individual dietary niche specialization and species environmental niches, although not citing any literature on behavioral specialization which in my mind is an important body of literature when talking about movement and resource selection behavior. I in particular liked Figure 1 and the introduction explaining alternative hypothetically niche configurations. To that end I also appreciated the discussion how different niche configuration scenarios would be differentially affected by global change.

Although I really liked the premise of the paper I have several major comments. Several are related to shortcomings in the study design and the resulting conclusions. Another comment pertains the confusion of behavioral plasticity versus individual variation which in my mind needs to be defined better in the paper. The biggest shortcoming of the paper is that it ignores the current state of the literature on individual variation in habitat use. As such it's claim of novelty is exaggerated since several papers have looked at individual specialization in habitat use, specialist to generalist gradients and variation in environmental niches in general habitat use or foraging behavior using movement data, some using similar methodology as used in this paper (Bastille-Rousseau et al. 2019). This existing body of literature must be acknowledged, cited appropriately and will help set the current study into context.

Here are a few seminal papers to start with but there are more:

Bastille-Rousseau, G. and G. Wittemyer, Leveraging multidimensional heterogeneity in resource selection to define movement tactics of animals. *Ecology Letters*, 2019. 22(9): p. 1417-1427.

(Looking at individual variation in habitat selection using movement data, RSF and clustering of individual RSF coefficients, analyses individual variation along different environmental covariates as behavioral hypervolumes)

Abrahms, B., et al., Climate mediates the success of migration strategies in a marine predator. *Ecology Letters*, 2018. 21(1): p. 63-71.

(A specialist to generalist gradient in site fidelity linked to climate conditions and forage intake)

Courbin, N., et al., Short-term prey field lability constrains individual specialisation in resource selection and foraging site fidelity in a marine predator. *Ecology Letters*, 2018. 21(7): p. 1043-1054.
(Variation in forage specialization as inferred from stable isotope linked to individual specialization in resource selection)

Leclerc, M., et al., Quantifying consistent individual differences in habitat selection. *Oecologia*, 2016. 180(3): p. 697-705.
(To my knowledge first paper assessing individual variation in habitat selection using movement data and RSF)

Harris, S.M., et al., Personality predicts foraging site fidelity and trip repeatability in a marine predator. *Journal of Animal Ecology*, 2020. 89(1): p. 68-79.
(Individual variation in foraging site fidelity from more repeatable (specialized) to less repeatable (generalist) individuals using movement data, linked to personality traits as discussed in this manuscript)

The implication of niche variation in the light of global change:

The authors make a strong case that individual niche specialization may play an important role in determining how resilient populations will be towards global change, a point that I would full heartedly endorse. I particularly liked how the authors outlined how different scenarios of niche configurations (Figure 1) would be differentially affected (line 227 - 238).

However, their empirical example unfortunately does not demonstrate any consequences of variation in niche space. Elsewhere in the literature, specialist versus generalist strategies have been linked to climate conditions with demonstrated effects on forage intake but this literature has not been cited (Abrahms et al. 2018). I again find the confusion of behavioral plasticity and phenotypic difference as underlying causes of specialization (line 231 -235) difficult. In my mind a specialist individual has a) limited plasticity over changing conditions and b) is highly repeatable in its behavior within one context while a generalist has high plasticity and may be additionally less repeatable under stable conditions.

Statistical considerations:

A shortcoming of the methodological approach is that only mean estimates of SSF models are taken forward into the niche model. The authors are thereby ignoring uncertainty around the selection estimates. Not taking forward uncertainty propagates error and potentially inflates niche estimates. I would suggest that the authors rerun their analysis using a bootstrap approach to account for uncertainty and to provide confidence intervals around niche estimates. See for example (Bastille-Rousseau and Wittemyer, 2019) with a different statistical approach but where uncertainty in resource selection coefficients is taken forward in downstream statistical models.

Study design:

If I understand it correctly, availability is not equal for all individuals but is sampled on a step by step basis (line 303): "Second, we evaluated these axes by performing step selection analysis, which reveals the individual selection of environmental conditions relative to availability."

Individual selection estimates are thus dependent on availability in the home range - if individual home ranges differ in availability this may directly affect selection (i.e. a functional response - a change in the selection of a habitat type depending on its availability (Myrsetrud and Ims, 1998)). The assumption that

the same resources are available to all individuals is not met. This is important because given the analytical approach it is unclear whether individuals are using a narrower range of habitats (a specialist environmental niche) because they only have a limited range of habitats available. This would indeed be a case of behavioral plasticity where individuals would show a similar environmental niche if they had the same resources available. It would be more convincing if all individuals would have the same habitat available but some would choose to use a narrower range whereas others use a wider range of environments. Because this study is done on wild animals, disentangling plasticity and true specialization is not trivial but there are statistical attempts how to do this (Hertel et al., 2020; Niemelä and Dingemanse, 2017; Sprau and Dingemanse, 2017).

This is an important point which is also mirrored in the discussion line 170: For example, strong selection for an environmental variable is indicative of specialist behavior and should result in narrower niche breadth. Alternatively, weaker selection, which is indicative of generalist behavior, means an animal is more likely to accept environments in proportion to their availability, thus resulting in relatively broad niche widths – this statement is only true if availability is the same for all individuals which does not seem to be the case here (again see (Mysterud and Ims, 1998)).

Environmental variables:

Though in the introduction distinct habitat types are discussed as providing alternative foraging opportunities (line 261 The white stork is an excellent species to investigate specialization because the species has characteristics that should result in populations composed of specialized individuals. White storks are generalist predators that forage in many different habitat types. Common prey items include worms in wet fields, frogs and fish in shallow ponds, mice, and voles and orthopterans in pasture and meadows. And line 312: White storks forage in a wide range of open habitats and avoid densely forested habitats. Foraging habitats include meadows, pasture, wetlands, riparian areas, recently plowed or harvested fields, and open woodlands.)

In the methodology surrogate variables are used (e.g. percent cover, distance to urban). I wonder whether these surrogate variables have a high accuracy in predicting the aforementioned distinct habitat types. I would expect that a stork that is specialized on a certain forage resource chooses for a specific habitat type. I wonder for example how well a pond would be predicted by the chosen covariates.

I believe that rerunning the analysis with habitat types instead of environmental covariates could be insightful and potentially yield stronger results. I understand if this is not possible due to a lack of regularly updated landcover maps (e.g. describing harvested fields) though a supervised classification of satellite images could produce such maps. The disparity between the chosen environmental covariates and the distinct habitat types (with distinct forage resources) should be addressed more in discussion.

Repeatability versus plasticity:

Along those lines (Line 388): “Repeatability in a particular trait implies reduced plasticity over different contexts”

This is not true, repeatability describes the amount of among individual variance to total variance (among and within individual variance) if R is high, individuals differ consistently in their behavior but this should not be confused with limited plasticity, individuals may differ in plasticity over different contexts (from individuals with limited to individuals with high plasticity towards changing context/environmental conditions). In this study, context did not vary, the authors chose one particular

period of the year. Variation in plasticity would be tested if habitat use in distinct periods, say spring and autumn, would be compared (using random slope models (Dingemanse et al., 2010)).

In my opinion individual variation in resource selection is also wrongfully interpreted as limited plasticity in the discussion line 193 – 195 Individuals showed consistent differences in resource selection and niche specialization over the four-year study period, implying limited plasticity in their response to the environment.

I believe what would be most interesting to show is whether niche breadth is repeatable while home range composition or location changes over years, if this was the case the authors indeed would have a strong case of niche specialization. If however a) storks simply show strong site fidelity in the home range they occupy over consecutive years, and b) the home range composition is relatively stable over years, then a repeatable niche breadth would be simply a byproduct of strong site fidelity and access to the same resources.

General comment:

Though the authors demonstrate that individuals differ in niche breadth and that niche breadth is repeatable over consecutive years, a further exploration into the drivers or consequences of this individual variation are lacking. The authors claim in the methods that age or size would not affect resource access and home range formation but I believe it would still be interesting (and necessary) to demonstrate that age, sex, or home range tenure do not affect niche breadth. (line 164 White storks do not have age or size structure in access to resources, and are not strictly territorial but often have overlapping home ranges. They will at times forage together in the same patch but can also display local aggression, chasing conspecifics away from their foraging area or nest.).

Another very obvious driver could be home range size. Bigger home ranges potentially hold a wider variety of resources (different habitats, wider variety of environmental conditions). Since availability was sampled on the home range scale (see comment on study design) a simple correlation whether home range size and niche breadth are correlated could shed light into this potential confound. If home ranges are approximately equal in size and you still see this variation in niche breadth, this is a big strength of the study that should be highlighted.

I wish you good luck in revising this timely manuscript

Anne Hertel

Cited references:

Abrahms B, Hazen EL, Bograd SJ, Brashares JS, Robinson PW, Scales KL, Crocker DE, Costa DP, Buckley L, 2018. Climate mediates the success of migration strategies in a marine predator. *Ecology Letters* 21:63-71. doi: <https://doi.org/10.1111/ele.12871>.

Bastille-Rousseau G, Wittemyer G, 2019. Leveraging multidimensional heterogeneity in resource selection to define movement tactics of animals. *Ecology Letters* 22:1417-1427. doi: [10.1111/ele.13327](https://doi.org/10.1111/ele.13327).

Courbin N, Besnard A, Péron C, Saraux C, Fort J, Perret S, Tornos J, Grémillet D, 2018. Short-term prey field lability constrains individual specialisation in resource selection and foraging site fidelity in a marine

predator. *Ecology Letters* 21:1043-1054. doi: <https://doi.org/10.1111/ele.12970>.

Dingemanse NJ, Kazem AJN, Réale D, Wright J, 2010. Behavioural reaction norms: animal personality meets individual plasticity. *Trends Ecol Evol* 25:81-89. doi: <http://dx.doi.org/10.1016/j.tree.2009.07.013>.

Harris SM, Descamps S, Sneddon LU, Bertrand P, Chastel O, Patrick SC, 2020. Personality predicts foraging site fidelity and trip repeatability in a marine predator. *Journal of Animal Ecology* 89:68-79. doi: [10.1111/1365-2656.13106](https://doi.org/10.1111/1365-2656.13106).

Hertel AG, Niemelä PT, Dingemanse NJ, Mueller T, 2020. A guide for studying among-individual behavioral variation from movement data in the wild. *Movement Ecology* 8:30. doi: [10.1186/s40462-020-00216-8](https://doi.org/10.1186/s40462-020-00216-8).

Leclerc M, Vander Wal E, Zedrosser A, Swenson JE, Kindberg J, Pelletier F, 2016. Quantifying consistent individual differences in habitat selection. *Oecologia* 180:697-705. doi: <https://doi.org/10.1007/s00442-015-3500-6>.

Mysterud A, Ims RA, 1998. Functional responses in habitat use: Availability influences relative use in trade-off situations. *Ecology* 79:1435-1441. doi: [10.2307/176754](https://doi.org/10.2307/176754).

Niemelä PT, Dingemanse NJ, 2017. Individual versus pseudo-repeatability in behaviour: Lessons from translocation experiments in a wild insect. *Journal of Animal Ecology* 86:1033-1043. doi: [10.1111/1365-2656.12688](https://doi.org/10.1111/1365-2656.12688).

Sprau P, Dingemanse NJ, 2017. An Approach to Distinguish between Plasticity and Non-random Distributions of Behavioral Types Along Urban Gradients in a Wild Passerine Bird. *Frontiers in Ecology and Evolution* 5. doi: [10.3389/fevo.2017.00092](https://doi.org/10.3389/fevo.2017.00092).

Reviewer #2 (Remarks to the Author):

The manuscript by Carlson and colleagues seeks to estimate the “Grinnellian” niche for 45 individual white storks from 3 populations in Germany, in order to understand individual-level niche specialization and consistency. It’s a rather interesting study, and well written, that uses some technologically advanced tracking devices to answer questions about used and experienced environments.

Despite these admirable qualities, I felt that the authors over-extended their results. To summarize (more below), it is unclear whether the Grinnellian niche is being measured in this study, and a lack of clear distinction makes it uncertain whether we should even expect that specialization in the Eltonian niche (which seems to occur in this species) may ever not result in apparent specialization of the Grinnellian niche (at least as measured in this study). As a result, the conclusions seem possibly pre-

ordained. Additionally, I found the evidence for niche stability to be weak and unconvincing, which further erodes my confidence that the study is actually measuring “niches” of individuals. This could be due to the lack of a rigorous framework for determining resource selection.

Framing:

What is the theory and justification behind the idea of an “individual niche”, as distinct from specialization? Does such a thing even exist? What would be necessary to define that an individual has a “niche”? I feel that the authors assume that an environmental niche is simply the environment experienced by an individual during its lifetime, but based on the underlying century of theory on a species’ ‘niche’, I’m not sure this simple assumption holds. Greater clarity would be appreciated.

Additionally, are we even sure that the ‘Grinnellian’ niche is being measured here? What is the distinction between the Eltonian niche for individuals, and the Grinnellian niche, as measured here? Based on the variables measured, which mostly relate to habitat usage (not, the “environment”, broadly speaking), it seems that these Grinnellian niches are arising because of Eltonian differentiation. That seems a bit different than the original formulation of Grinnellian versus Eltonian niches, which are assumed to be more or less independent of each other (e.g., two sympatric species can occupy the same Grinnellian niche but very different Eltonian niches). I suppose my question is, are you actually measuring the Grinnellian niche here, or are you measuring the Eltonian niche as seen through landscape variables? This relates to my question above, about the lack of a stronger theoretical foundation for the current study.

The question about niche configuration is very interesting (Fig 1), although I doubt that all 4 configurations exist clearly in the wild. The first two questions, however (lines 107-109) are highly exploratory, rather than hypothesis-driven. Don’t we already expect that we should be able to detect individual variation in environmental preferences? Is there any way that someone could study 49 individuals and not detect individual variation in experienced environments? Is there a null model here? The second question is about how narrow individual niches are relative to population niches. We know that individuals likely won’t experience the full range of environmental or landscape conditions, but is there a prior expectation of how narrow or wide would be surprising? (more on this below, in approach.)

Approach:

Fundamentally, I feel the analysis suffers from a lack of null models for determining what would be expected, at random. For example, if use points were randomly assigned to individuals, and the key metrics were re-calculated (e.g., nestedness, etc.), how much ‘random’ stability would be expected? Re-sampling methods could be extremely helpful in better assessing what is surprising (or not surprising) in these results, for all questions.

As framed, this study aims to connect individual environmental variation as it scales within “one species”, yet it focuses only on 45 individuals from 3 populations (in one season) of a species that has a very wide range and distribution. Can the results and their interpretation actually scale from individuals

to a species, as suggested in the introduction? Or, more accurately, is this analysis only scaling from individuals to a population? And if so, what are we missing from not being able to scale from the population level to the species level?

Finally, a key aspect of any resource selection analysis is the choice of 'use' versus 'available' points. While the choice of 'use' points in this study are very well justified (based on accelerometer data), I am much more dubious about how 'available' points are chosen, particularly as the selection of 'available' points can have extremely strong influences on the results of any selection analysis. What perhaps is more worrisome is that the methods never specify how 'available' points were chosen. They imply, however, that available points were chosen randomly from all habitats within a certain radius of a centroid of the population. Such an approach would have several problems, however: it assumes that all habitats are equally accessible among individuals and it assumes that all habitats are suitable for use even when clearly not (e.g. if 'use' is foraging, then an asphalt parking lot cannot be 'available'). It seems that for any analysis of 'individual niches,' that availability needs to be defined individually, otherwise simple aspects of spatial segregation and home range limitations (i.e., the random distribution of individual on a landscape) can lead to the strong appearance of "specialization" in niches.

Interpretation:

The only result I don't easily see in the data is the final result, that niches are stable over time. Figure 4 appears to show a huge amount of interannual variation within individuals, as well as temporal trends (e.g. in specialization) at the population level, which further indicates non-stability. The authors will have to do a more convincing job to demonstrate that these niches are stable and consistent, rather than arbitrary. Null models (see above), would help.

The application of the results, as presented, to a broad critique of SDMs and how they are used in global change research seems like a stretch and is beyond the bounds of the research findings. I don't disagree that the individual-level behavior may impact SDM performance, but the current analysis sheds limited light on how much of a problem this might actually be, particularly as SDMs are often done at the species level, not the population level (like this study).

Additional:

Methods Disclaimer says that code is available from author upon request. As authors often move institutions and may not be easily tracked down, all code should be deposited on a public repository, such as GitHub and linked within the data accessibility page.

For sampling strategy, it says that storks were haphazardly sampled. Were storks sampled preferentially with respect to age or sex?

Reviewer #1

Overview

The manuscript entitled „Individual environmental niches in mobile organisms“ by Carlson et al. uses movement data of 45 white storks to expand the concept of intra-specific variation in dietary niche breadth and composition (Eltonian niche) to intra-specific variation in environmental niche breadth and composition (Grinellian niche). Traditionally, grinellian niches are typically estimated on the species level using species distribution models (SDM's).

The authors here use resource selection functions (more specifically, step selection functions SSF) fit on individuals to show that storks differ in habitat selection with some storks using a wider range of habitats than others.

The authors conclude that intraspecific variation in environmental niches exist and that white storks can be organized along a specialist to generalist gradient with all storks using a core of the same environment but some additionally also use other environments.

I generally agree with the premise of the paper and congratulate the authors for their elegant conceptual approach of combining the literature on individual dietary niche specialization and species environmental niches, although not citing any literature on behavioral specialization which in my mind is an important body of literature when talking about movement and resource selection behavior.

I in particular liked Figure 1 and the introduction explaining alternative hypothetically niche configurations. To that end I also appreciated the discussion how different niche configuration scenarios would be differentially affected by global change.

We thank the reviewer for their careful assessment of our work and are pleased they appreciate our novel conceptual foundation and links we develop to global change. We should note that resource selection functions were used as methodological step in order to identify relevant hypervolumes axes, which in turn enabled key insights around niche breadth and specialization, but otherwise were not a core focus of our study. Please see below for additional discussion. We recognize the limited citations addressing behavioral specialization and fully agree with the reviewer's suggestion here. We have now added text in several places citing this literature and comparing/contrasting this body of literature with our study.

Although I really liked the premise of the paper I have several major comments. Several are related to shortcomings in the study design and the resulting conclusions. Another comment pertains the confusion of behavioral plasticity versus individual variation which in my mind needs to be defined better in the paper.

Comment I:

The biggest shortcoming of the paper is that it ignores the current state of the literature on individual variation in habitat use. As such its claim of novelty is exaggerated since several papers have looked at individual specialization in habitat use, specialist to generalist gradients and variation in environmental niches in general habitat use or foraging behavior using movement data, some using similar methodology as used in this paper (Bastille-Rousseau et al. 2019). This existing body of literature must be acknowledged, cited appropriately and will help set the current study into context.

Here are a few seminal papers to start with but there are more:

Bastille-Rousseau, G. and G. Wittemyer, Leveraging multidimensional heterogeneity in resource selection to define movement tactics of animals. *Ecology Letters*, 2019. 22(9): p. 1417-1427.
(Looking at individual variation in habitat selection using movement data, RSF and clustering of individual RSF coefficients, analyses individual variation along different environmental covariates as behavioral hypervolumes)

Abrahms, B., et al., Climate mediates the success of migration strategies in a marine predator. *Ecology Letters*, 2018. 21(1): p. 63-71.
(A specialist to generalist gradient in site fidelity linked to climate conditions and forage intake)

Courbin, N., et al., Short-term prey field lability constrains individual specialisation in resource selection and foraging site fidelity in a marine predator. *Ecology Letters*, 2018. 21(7): p. 1043-1054.
(Variation in forage specialization as inferred from stable isotope linked to individual specialization in resource selection)

Leclerc, M., et al., Quantifying consistent individual differences in habitat selection. *Oecologia*, 2016. 180(3): p. 697-705.
(To my knowledge first paper assessing individual variation in habitat selection using movement data and RSF)

Harris, S.M., et al., Personality predicts foraging site fidelity and trip repeatability in a marine predator. *Journal of Animal Ecology*, 2020. 89(1): p. 68-79.
(Individual variation in foraging site fidelity from more repeatable (specialized) to less repeatable (generalist) individuals using movement data, linked to personality traits as discussed in this manuscript)

Comment 1 – Response

We fully agree that there are several significant studies addressing individual variation in behaviors such as habitat selection. It was not our intention to imply that this aspect of our study was novel. Instead, we feel that the most novel aspect of our study is in our representation of multivariate environmental niches as hypervolumes, which allows us to understand geometric relationships among individual usage distributions. To our knowledge,

studies of individual variation in habitat use (including those studies above in the reviewer's very helpful list) generally examine variation in behaviors, but not in multi-dimensional use of environments. For example, work using resource selection functions (RSFs) generally addresses variation in habitat selection, a type of behavior (e.g. Beyer *et al.*, 2010). Although RSF/SSF are sometimes referred to as "niches" we agree with others who maintain that the niche concept should be reserved for objects that directly estimate usage in niche space, rather than for indices of behavior. For example, Matthiopolous et al. (2020) writes:

"*E*-space can be considered identical to niche space, as originally conceived by Hutchinson (1957) and MacArthur (1968), although, as extensively argued in the modern literature (Soberón and Nakamura 2009; Peterson et al. 2011; McInerny and Etienne 2013; Matthiopoulos et al. 2015), statistical habitat preference models currently fitted in *E*-space should not be confused with the niche objects as envisaged by these pioneering thinkers".

Bastille-Rousseau et al. (2019) is a good illustration of the differences between analyses of behavior and usage distributions. In this study, the authors use RSF analysis to estimate habitat selection preferences. They then use these indices of selection in a cluster analysis to identify similarity in behavioral strategies among individuals. They use the concept of a behavioral hypervolume to describe the collective behavioral strategies of a population, and specifically contrast this against the ecological hypervolume (or niche) in the traditional sense of Hutchinson. We note that Bastille-Rousseau did not claim to estimate ecological hypervolumes. This is a fundamental difference between our study and theirs. We estimate a specific type of ecological hypervolume, one composed of environmental variables: the Grinnellian niche *sensu* Soberon et al. (2009). Thus, we are able to test hypotheses of niche configuration that are not possible using the approach in Bastille-Rousseau et al.. For example, the RSF estimates from Bastille-Rousseau et al. represent single points in regression parameter space, but as single points lack geometric properties such as size or shape. This means they cannot be used to examine geometric relationships such as the level of specialization or nestedness. In addition, the RSF estimates from Bastille-Rousseau do not directly describe the usage distribution in environmental space, but instead index the probability that an individual will select a particular level of that variable, relative to the level of that variable available to it. Although this is a very useful quantity to estimate, resource selection represents an indirect measure of usage, mediated by a behavioral parameter. We believe that directly representing environmental usage as hypervolumes (niches) provides an alternative and important way to understand how individual animals use the environment.

Thus, we believe that the novel aspect of our study is that we are, to our knowledge, the first to represent niches as individual hypervolumes based on usage distributions, in the way that Hutchinson would have intended. Using hypervolumes to examine variation in environmental usage allows us to test hypotheses of niche configuration such as specialization, nestedness, and clustering that are not possible using a behavioral approach. The specialization literature in the tradition of Bolnick et al. (2003) also focuses on usage, but has thus far focused on Eltonian variables and usually does not examine inter-relationships among individual niches (configuration). Studies of environmental use generally focus on identifying variation in

behavior, but not on usage distributions in environmental space. By employing individual hypervolumes based on usage distributions, we extend analyses based on the seminal ideas of Hutchinson (1957, 1978) and in the tradition of Bolnick et al. (2003) to individual Grinnellian niches.

We have updated the text to further clarify this distinction and novelty, and we have incorporated the requested citations (plus additional references) to provide additional context to our study. We are of course fully open to hearing how we could do even better here.

Comment 2:

The implication of niche variation in the light of global change:

The authors make a strong case that individual niche specialization may play an important role in determining how resilient populations will be towards global change, a point that I would full heartedly endorse. I in particular liked how the authors outlined how different scenarios of niche configurations (Figure 1) would be differentially affected (line 227 - 238).

However, their empirical example unfortunately does not demonstrate any consequences of variation in niche space.

Comment 2 - Response:

We appreciate the reviewer's recognition that our study makes a strong case for the role of individual niche specialization in climate change resilience. It is correct that we did not include a quantitative assessment of projected consequences of climate change for these populations. We believe a rigorous demonstration requires a full and quantitative assessment that should ideally include a range of fine-scale change projections for different climate scenarios. As is, our study provides an extension of niche specialization into Grinnellian space, the novel use of hypervolumes to represent individual niches, and identification of surprising patterns of niche configuration (compared to studies of Eltonian niches). Given the breadth and length of the current manuscript, specific climate change projections might therefore be best addressed in a separate study. But we are fully open to specific suggestions from the reviewer or editor regarding any extensions. As currently phrased, we are not sure whether the reviewer firmly requests such an addition (and in which form).

Comment 3:

Elsewhere in the literature, specialist versus generalist strategies have been linked to climate conditions with demonstrated effects on forage intake but this literature has not been cited (Abrahms et al. 2018).

Comment 3 - response:

Thank you for pointing out this study. We now refer to this study in the text.

Comment 4:

I again find the confusion of behavioral plasticity and phenotypic difference as underlying causes of specialization (line 231 -235) difficult. In my mind a specialist individual has a) limited plasticity over changing conditions and b) is highly repeatable in its behavior within one context while a generalist has high plasticity and may be additionally less repeatable under stable conditions.

Because, as you rightly point out, we do not explicitly test for individual plasticity, we have focused these discussion points on consistent among individual differences. Please see below for further discussion.

However, we feel it is worth mentioning that the concept of niche specialization in the tradition of Bolnick et al. (2003) is not the same as the definition you describe above. Niche specialization is usually considered context dependent and can be plastic from one context to the next. An often-hypothesized reason has to do with the level of competition. When competition is low, individuals are all thought to focus on the same high profitability (high-energy, low handling time) prey. Thus, all individuals have similar niches. When competition increases, perhaps due to higher density and/or exhaustion of the best resource, individuals are forced to find alternative prey. In this scenario, individuals focus on different prey items (Svanback & Bolnick 2005). Niche expansion occurs, but the mechanism of expansion is niche specialization among individuals. Thus, the level of specialization can change in response to ecological context. This is different from concepts in personality analysis, in which there is a focus on individual consistency among contexts, and an individual demonstrating low plasticity among contexts might be called a specialist. It is unfortunate that “specialist” individuals, in the sense of limited plasticity over different contexts, and “specialization” in the sense of narrow individual niches relative to population niches are different concepts but have similar names. There have been limited attempts to merge the fields of niche specialization in the tradition of Bolnick et. al (2003) and personality research. One notable attempt is Toscano et al. (2016).

Statistical considerations:

A shortcoming of the methodological approach is that only mean estimates of SSF models are taken forward into the niche model. The authors are thereby ignoring uncertainty around the selection estimates. Not taking forward uncertainty propagates error and potentially inflates niche estimates. I would suggest that the authors rerun their analysis using a bootstrap approach to account for uncertainty and to provide confidence intervals around niche estimates. See for example (Bastille-Rousseau and Wittemyer, 2019) with a different statistical approach but where uncertainty in resource selection coefficients is taken forward in downstream statistical models.

We thank you for your point about our lack of uncertainty in niche estimates. As suggested, we performed bootstrap analysis to estimate uncertainty for individual niche volume and report these results in Supplementary File S1. In addition, we use a bootstrapping approach to test two hypotheses 1) that individual identity does not play a role in the size of the niche metrics, and 2) that available background does not play a role in the value of the niche metrics. Please see below for further information.

We also feel we should point out that it is not accurate to state that mean SSF estimates were taken forward into the niche model. In fact, we do not use the SSF estimates at all in the hypervolume analysis. The SSF analysis is simply used as a methodological step to rigorously evaluate our choice of environmental variables. We interpreted statistically significant selection (positive or negative) as evidence that the environmental variable was important to the individual and was thus likely a niche axis. Based on this step, we chose to use the environmental variables in our hypervolume analysis. Other than affording us binary yes/no decision, the SSF analysis did not factor any further into our niche (hypervolume) analysis. It is an interesting idea to integrate SSF analysis with hypervolume analysis as you describe. But, to our knowledge this has never been attempted and first require a range of additional methodological developments.

We made several changes in order to better describe our use of hypervolumes, including paragraphs and text to the introduction, methods and results. In addition, we moved Fig. 2, which showed SSF results, to the supplement in order to clearly show that the SSF analysis is not the core part of this study.

Study design:

If I understand it correctly, availability is not equal for all individuals but is sampled on a step by step basis (line 303): “Second, we evaluated these axes by performing step selection analysis, which reveals the individual selection of environmental conditions relative to availability.” Individual selection estimates are thus dependent on availability in the home range - if individual home ranges differ in availability this may directly affect selection (i.e. a functional response - a change in the selection of a habitat type depending on its availability (Mysterud and Ims, 1998)).

The assumption that the same resources are available to all individuals is not met. This is important because given the analytical approach it is unclear whether individuals are using a narrower range of habitats (a specialist environmental niche) because they only have a limited range of habitats available. This would indeed be a case of behavioral plasticity where individuals would show a similar environmental niche if they had the same resources available. It would be more convincing if all individuals would have the same habitat available but some would choose to use a narrower range whereas others use a wider range of environments. Because this study is done on wild animals, disentangling plasticity and true specialization is not trivial but there are statistical attempts

how to do this (Hertel et al., 2020; Niemelä and Dingemanse, 2017; Sprau and Dingemanse, 2017).

This is an important which is also mirrored in the discussion line 170: For example, strong selection for an environmental variable is indicative of specialist behavior and should result in narrower niche breadth. Alternatively, weaker selection, which is indicative of generalist behavior, means an animal is more likely to accept environments in proportion to their availability, thus resulting in relatively broad niche widths – this statement is only true if availability is the same for all individuals which does not seem to be the case here (again see (Mysterud and Ims, 1998)).

Thank you for pointing out this important issue. We agree that further analysis of null models, including available backgrounds, is warranted and have performed additional analyses in which we used a bootstrapping approach to test two separate null models.

We tested the null model that individual identity did not play a role in the value of the metrics. We performed this test by ignoring identity and sampling from the full population, then computing niche metrics. We repeated this procedure one hundred times to see the distributions of niche metrics. The results (S3) show that individual identity appears to play a strong role in the value of specialization and nestedness metrics, but clustering does not appear different from the null expectation. However, we note that the underlying mechanism for the role of individual identity could still be differences in the habitats available to individuals, due to the placement of individuals nests and home ranges on the landscape. To test this, we performed additional boot-strapping on the backgrounds available to each individual.

We tested the hypothesis that our values of niche metrics were due to differential background availability by sampling backgrounds based on the space-use of each individual every two weeks, in order to account for constraints due to breeding phenology (please see the methods section for additional details). We sampled from these backgrounds, and calculated niche metrics using these data, using a bootstrap approach to estimate the sampling distributions. Our results (S3) show that while the available background plays a larger role in the level of specialization than the scenario in which individual identity is not important, in many cases there is still additional variation due to individual selection of habitats within an individual's respective background. In particular, 9/12 of the population/years would reject the hypothesis that specialization is due to available environment with $p < 0.05$, with one additional population/year moderately significant with $p = 0.06$. Nestedness does not appear to be driven by the available background to any higher degree than lack of individual identity. Similar to the scenario where individual identity is not important, storks also did not depart from the null distribution for clustering.

To obtain a value of clustering < 1 , at least two (or more) niches need to be disconnected from each other. That is, they need to have Jaccard overlap < 0.05 . It is hypothetically possible that individual animals partition environmental space to such a degree that there is very little overlap in their niches, but we don't find that this is true with our populations. Thus, we don't

find any evidence for modularity in our study. We have included this explanation when discussing our results.

Environmental variables:

Though in the introduction distinct habitat types are discussed as providing alternative foraging opportunities (line 261 The white stork is an excellent species to investigate specialization because the species has characteristics that should result in populations composed of specialized individuals. White storks are generalist predators that forage in many different habitat types. Common prey items include worms in wet fields, frogs and fish in shallow ponds, mice, and voles and orthopterans in pasture and meadows. And line 312: White storks forage in a wide range of open habitats and avoid densely forested habitats. Foraging habitats include meadows, pasture, wetlands, riparian areas, recently plowed or harvested fields, and open woodlands.)

In the methodology surrogate variables are used (e.g. percent cover, distance to urban). I wonder whether these surrogate variables have a high accuracy in predicting the aforementioned distinct habitat types. I would expect that a stork that is specialized on a certain forage resource chooses for a specific habitat type. I wonder for example how well a pond would be predicted by the chosen covariates.

I believe that rerunning the analysis with habitat types instead of environmental covariates could be insightful and potentially yield stronger results. I understand if this is not possible due to a lack of regularly updated landcover maps (e.g. describing harvested fields) though a supervised classification of satellite images could produce such maps. The disparity between the chosen environmental covariates and the distinct habitat types (with distinct forage resources) should be addressed more in discussion.

Thank you for pointing out this disconnect in how we describe stork habitat and the variables we use. In fact, several references show that individual white storks may have individual variation in the variables we have chosen, but we did not explicitly discuss this in the study. We have updated the methods section to more directly discuss the habitat variables we used.

Repeatability versus plasticity:

Along those lines (Line 388): "Repeatability in a particular trait implies reduced plasticity over different contexts"

This is not true, repeatability describes the amount of among individual variance to total variance (among and within individual variance) if R is high, individuals differ consistently in their behavior but this should not be confused with limited plasticity, individuals may differ in plasticity over different contexts (from individuals with limited to individuals with high plasticity towards changing context/environmental conditions).

In this study, context did not vary, the authors chose one particular period of the year. Variation in plasticity would be tested if habitat use in distinct periods, say spring and autumn, would be compared (using random slope models (Dingemanse et al., 2010)).

In my opinion individual variation in resource selection is also wrongfully interpreted as limited plasticity in the discussion line 193 – 195 Individuals showed consistent differences in resource selection and niche specialization over the four-year study period, implying limited plasticity in their response to the environment.

Thank you for pointing out our imprecise use of plasticity. We agree that we did not explicitly test for plasticity in the sense of reaction norms as discussed in Dingamane et al. (2010). Because we measure repeatability over time (albeit in the same season each year), we have instead updated the text to discuss consistent among-individual differences.

I believe what would be most interesting to show is whether niche breadth is repeatable while home range composition or location changes over years, if this was the case the authors indeed would have a strong case of niche specialization. If however a) storks simply show strong site fidelity in the home range they occupy over consecutive years, and b) the home range composition is relatively stable over years, then a repeatable niche breadth would be simply a byproduct of strong site fidelity and access to the same resources.

This is an excellent suggestion, although the consistency of individual nest site selection would not provide us with the power to disentangle the role of stability in home range habitat composition in driving repeatability in our populations. However, we note that our bootstrap analysis shows that not all specialization is due to the available background. In addition, there are alternative hypotheses for why individuals might show consistent individual differences across years. For example, if the level of competition was similar across years, this might also constrain individual niche breadth. There is some evidence to back up this claim, since it is thought that stork population growth during the breeding period tends to be limited by density-independent factors (such as cold rain events in early spring). However, it was not our goal to explicitly test these hypotheses, but only to point them out in the discussion. This could be a fruitful and interesting area of future research.

General comment:

Though the authors demonstrate that individuals differ in niche breadth and that niche breadth is repeatable over consecutive years, a further exploration into the drivers or consequences of this individual variation are lacking.

We feel that these are excellent topics for future research, but are outside the scope of this study. We now mention this opportunity in the discussion.

The authors claim in the methods that age or size would not affect resource access and home range formation but I believe it would still be interesting (and necessary) to demonstrate that age, sex, or home range tenure do not affect niche breadth. (line 164 White storks do not have age or size structure in access to resources, and are not strictly territorial but often have

overlapping home ranges. They will at times forage together in the same patch but can also display local aggression, chasing conspecifics away from their foraging area or nest.). Another very obvious driver could be home range size. Bigger home ranges potentially hold a wider variety of resources (different habitats, wider variety of environmental conditions). Since availability was sampled on the home range scale (see comment on study design) a simple correlation whether home range size and niche breadth are correlated could shed light into this potential confound. If home ranges are approximately equal in size and you still see this variation in niche breadth, this is a big strength of the study that should be highlighted.

We have now included two additional figures in the supplement. We show that niche breadth is not a function of sex (Fig. S5) and is not strongly driven by home range size (Fig. S4). Although there is a small correlation between niche breadth and home range size ($r=0.3$, adjusted $r^2=0.11$), this relationship does not appear to be a strong driver of individual niche differences.

I wish you good luck in revising this timely manuscript
Anne Hertel

Cited references:

Abrahms B, Hazen EL, Bograd SJ, Brashares JS, Robinson PW, Scales KL, Crocker DE, Costa DP, Buckley L, 2018. Climate mediates the success of migration strategies in a marine predator. Ecology Letters 21:63-71. doi: <https://doi.org/10.1111/ele.12871>.

Bastille-Rousseau G, Wittemyer G, 2019. Leveraging multidimensional heterogeneity in resource selection to define movement tactics of animals. Ecology Letters 22:1417-1427. doi: 10.1111/ele.13327.

Courbin N, Besnard A, Péron C, Saraux C, Fort J, Perret S, Tornos J, Grémillet D, 2018. Short-term prey field lability constrains individual specialisation in resource selection and foraging site fidelity in a marine predator. Ecology Letters 21:1043-1054. doi: <https://doi.org/10.1111/ele.12970>.

Dingemanse NJ, Kazem AJN, Réale D, Wright J, 2010. Behavioural reaction norms: animal personality meets individual plasticity. Trends Ecol Evol 25:81-89. doi: <http://dx.doi.org/10.1016/j.tree.2009.07.013>.

Harris SM, Descamps S, Sneddon LU, Bertrand P, Chastel O, Patrick SC, 2020. Personality predicts foraging site fidelity and trip repeatability in a marine predator. Journal of Animal Ecology 89:68-79. doi: 10.1111/1365-2656.13106.

Hertel AG, Niemelä PT, Dingemanse NJ, Mueller T, 2020. A guide for studying among-individual behavioral variation from movement data in the wild. Movement Ecology 8:30. doi: 10.1186/s40462-020-00216-8.

Leclerc M, Vander Wal E, Zedrosser A, Swenson JE, Kindberg J, Pelletier F, 2016. Quantifying consistent individual differences in habitat selection. *Oecologia* 180:697-705. doi: <https://doi.org/10.1007/s00442-015-3500-6>.

Mysterud A, Ims RA, 1998. Functional responses in habitat use: Availability influences relative use in trade-off situations. *Ecology* 79:1435-1441. doi: 10.2307/176754.

Niemelä PT, Dingemanse NJ, 2017. Individual versus pseudo-repeatability in behaviour: Lessons from translocation experiments in a wild insect. *Journal of Animal Ecology* 86:1033-1043. doi: 10.1111/1365-2656.12688.

Sprau P, Dingemanse NJ, 2017. An Approach to Distinguish between Plasticity and Non-random Distributions of Behavioral Types Along Urban Gradients in a Wild Passerine Bird. *Frontiers in Ecology and Evolution* 5. doi: 10.3389/fevo.2017.00092.

Reviewer #2 (Remarks to the Author):

The manuscript by Carlson and colleagues seeks to estimate the “Grinnellian” niche for 45 individual white storks from 3 populations in Germany, in order to understand individual-level niche specialization and consistency. It’s a rather interesting study, and well written, that uses some technologically advanced tracking devices to answer questions about used and experienced environments.

Despite these admirable qualities, I felt that the authors over-extended their results. To summarize (more below), it is unclear whether the Grinnellian niche is being measured in this study, and a lack of clear distinction makes it uncertain whether we should even expect that specialization in the Eltonian niche (which seems to occur in this species) may ever not result in apparent specialization of the Grinnellian niche (at least as measured in this study). As a result, the conclusions seem possibly pre-ordained.

Additionally, I found the evidence for niche stability to be weak and unconvincing, which further erodes my confidence that the study is actually measuring “niches” of individuals. This could be due to the lack of a rigorous framework for determining resource selection.

We thank the reviewer for their thoughtful assessment and interest in the study.

We provide detailed discussion of this feedback below and a short summary here. Our approach to Grinnellian niche assessment is grounded in an approach that is, at species-level, broadly recognized and very widely used. Grinnellian and Eltonian niches are likely related by a complex set of factors, but irrespective of the exact mechanistic link between Eltonian and Grinnellian niche specialization (which is empirically hard to establish) the study is to our knowledge the first to conceptually lay out and measure Grinnellian niche specialization by representing Grinnellian individual niches as multivariate usage distributions (Hypervolumes) in the spirit of Hutchinson. To date, studies of niche specialization (e.g. Bolnick *et al.*, 2003)

examine usage distributions but are focused on Eltonian variables and univariate (or bi-variate) distributions and do not examine the inter-relationships among individual niches (niche configuration). In contrast, studies of Grinnellian variables adopt a multivariate approach but focus on behavioral indices (e.g. resource selection, step selection) and do not examine multivariate usage distributions. To our knowledge, our study is the first to integrate these two approaches. The vast environmental niche and species distribution modelling literature remains so far largely devoid of relevant conceptualizations and empirical demonstrations of the relevance of individual variation and specialization. Therein lies the key advance provided by our study.

Framing:

What is the theory and justification behind the idea of an “individual niche”, as distinct from specialization? Does such a thing even exist? What would be necessary to define that an individual has a “niche”? I feel that the authors assume that an environmental niche is simply the environment experienced by an individual during its lifetime, but based on the underlying century of theory on a species’ ‘niche’, I’m not sure this simple assumption holds. Greater clarity would be appreciated.

We thank the reviewer for these important and interesting philosophical questions. Few topics in ecology are as widely debated and have seen as varying and overlapping definitions as the “niche”. A full treatise of the topic and the interesting questions the reviewer raises are outside the scope of the study. However, we made significant revisions to the main text to further refine the concepts and definitions we used, and provide further reflections and responses for the reviewer in this response.

First, we follow several principles of modern thinking on niche concepts in our definition of an individual niche. One important principle follows Hutchinson’s (1957, 1978) definition of the niche as a hypervolume as well as the definition that niches are properties of species (or here, individuals), and not locations in the physical environment (Colwell & Rangel, 2009). This definition implies that no two species (or individuals) can have the same niche. This does not mean that two niches can’t be equivalent in the sense of having similar characteristics, but that each species (or in our case, individual) has its own fundamental niche, which is defined in terms of its phenotype. For example, two birds might have the same beak measurements, but we would still say they each have their own beak. Colwell & Rangel (2009) provides a well-written review of these concepts. Extending this concept to individuals, we further assume that population niches can be seen as the combination of all the individual niches of that population. Thus, we can consider the size of an individual’s niche compared to the size of its population’s niche. Taking the definition directly from Bolnick et al. (2003), when an individual’s niche is narrow relative to the population’s niche, we say that individual exhibits niche specialization (or that it is specialized). When, on average, individual population niches are narrow relative to the population’s niche we say that the population is composed of individual specialists, or that the population is specialized. Thus, our definition of individual niche and specialization are taken directly from foundational, generally agreed-upon concepts and well-established literature.

Surprisingly, despite hundreds of papers written on the topic of individual niches, to our knowledge there is not a formal definition of an individual niche *sensu* Hutchinson (1957, 1978). However, we adopt an operational definition by extending Hutchinson's concept and the principles described above to the level of the individual. Hutchinson's hypervolume concept states that a population (or species) niche is the regions of a hypervolume that permit positive density-independent growth. Although the concept of population growth does not apply to individuals, we extend Hutchinson's definition to individuals by considering an individual niche as regions in niche space that permit an individual to survive and reproduce. Thus, we do not simply consider any environmental conditions the individual experiences as part of its niche, but only those conditions that allow the individual to survive and reproduce. This is one reason why we focus on foraging locations during the breeding period in our study, and have filtered out other locations such as sitting, preening, flying, etc. By focusing on the environmental characteristics of an individual's foraging locations, which are closely tied to the individual's ability to survive and reproduce, we estimate individual realized niches.

Additionally, are we even sure that the 'Grinnellian' niche is being measured here? What is the distinction between the Eltonian niche for individuals, and the Grinnellian niche, as measured here? Based on the variables measured, which mostly relate to habitat usage (not, the "environment", broadly speaking), it seems that these Grinnellian niches are arising because of Eltonian differentiation. That seems a bit different than the original formulation of Grinnellian versus Eltonian niches, which are assumed to be more or less independent of each other (e.g., two sympatric species can occupy the same Grinnellian niche but very different Eltonian niches). I suppose my question is, are you actually measuring the Grinnellian niche here, or are you measuring the Eltonian niche as seen through landscape variables? This relates to my question above, about the lack of a stronger theoretical foundation for the current study.

Continuing our discussion from above, we also follow well-established concepts for our definition of Eltonian and Grinnellian niches. Here, we follow several seminal papers by Soberón et al. (2005; 2007, 2010; 2009) and a follow up book by Peterson et al. (2011) in our definition. These concepts hold that niche axes can be divided into two categories, in terms of how species (or individuals) interact with them. Variables whose levels are actively changed by the presence of an organism, such as prey abundance, are termed interactive variables. Those that are not actively changed by the organism (at least not on a time scale relative to the temporal grain of the study), such as land cover or coarse-grained temperature, are termed non-interactive variables. Combining these concepts with Hutchinson's definition of the niche, a hypervolume composed of interactive variables is called an Eltonian niche, and a hypervolume composed of non-interactive variables is called a Grinnellian niche. Note that these definitions are unfortunately not the same as the definitions that Grinnell and Elton originally proposed. This situation, can be confusing, leading some to call for abandoning "author-ian" niche names altogether (McInerney & Etienne 2012). However, the definitions by Soberón et al. are well established in the modern literature so we also use them here. All the definitions from the discussion to the previous question also apply, which means that each species (or individual)

has its own Eltonian and Grinnellian niche. Based on these definitions, it is easy to see that our study, which examines environmental axes (such as tree cover, or distance to urban) that are not changed by the presence of individuals (at the time scale of our study) estimate individual Grinnellian niches.

It is also straight-forward to see that Eltonian and Grinnellian niches are not independent but have some interaction with each other, although the relationship is likely complicated. In some cases, these niches might covary. For example, if individuals target a specific prey species, and that species is only found in particular habitats, the Eltonian niche might predict the Grinnellian niche. Or, if an individual instead focused on a particular type of habit, and will generally consume anything in that habitat, then the Grinnellian niche might predict the Eltonian niche. One could also imagine many scenarios in which this tight coupling between Eltonian and Grinnellian niches does not exist. It is not our intention to disentangle these relationships in this study. However, due to these potentially complicated relationships we feel that it is important to directly examine Grinnellian niches and not assume their patterns can be directly derived from Eltonian niches. In fact, our results show that for our populations, Grinnellian niche patterns may not follow Eltonian patterns. Almost all studies of Eltonian niches assume that individuals discretely partition niche space (e.g. Bolnick *et al.*, 2003; Araújo *et al.*, 2011; Costa-Pereira *et al.*, 2018), but we do not find this pattern in Grinnellian niches. Instead, we find that Grinnellian niches are nested.

We have added text into the introduction that discusses these points. In particular, we discuss why it is important to assess variation in Grinnellian niche patterns directly and why Eltonian niche configuration patterns might not predict Grinnellian patterns.

The question about niche configuration is very interesting (Fig 1), although I doubt that all 4 configurations exist clearly in the wild. The first two questions, however (lines 107-109) are highly exploratory, rather than hypothesis-driven. Don't we already expect that we should be able to detect individual variation in environmental preferences? Is there any way that someone could study 49 individuals and not detect individual variation in experienced environments? Is there a null model here? The second question is about how narrow individual niches are relative to population niches. We know that individuals likely won't experience the full range of environmental or landscape conditions, but is there a prior expectation of how narrow or wide would be surprising? (more on this below, in approach.)

Thank you, we have updated our questions to make them more hypothesis-driven. We appreciate the feedback that the answer to the first two (original) questions may both be obvious and 'yes', but we think that would be making hasty conclusions on very little evidence. We are not aware of any literature or evidence that examines variation in individual Grinnellian niches. Thus, we feel it is important to establish quantitative evidence supporting the answers to these questions.

We fully agree, and appreciate the feedback, that further analysis of null models, including available backgrounds, would be beneficial to better understand the drivers of individual

variation. We used a bootstrapping approach to test two separate null models as described below.

We tested the null model that individual identity did not play a role in the value of the metrics. We performed this test by ignoring identity and sampling from the full population, then computing niche metrics. We repeated this procedure one hundred times to see the distributions of niche metrics. The results (S3) show that individual identity appears to play a strong role in the value of specialization and nestedness metrics, but clustering does not appear different from the null expectation. However, we note that the underlying driver for the importance of individual identity could still be differences in the habitats available to individuals due to the placement of individuals nests and home ranges on the landscape. To test this we performed additional boot-strapping on the backgrounds available to each individual.

We tested the hypothesis that niche metrics were due to differential background availability by sampling backgrounds based on the space-use of each individual every two weeks, in order to account for constraints due to breeding phenology (please see the methods section for additional details). We sampled from these backgrounds, and calculated niche metrics using these data, using a bootstrap approach to estimate the sampling distributions. Our results (S3) show that while the available background plays a larger role in the level of specialization than the scenario in which individual identity is not important, in many cases there is still additional variation due to individual selection habitats within individual's respective background. In particular, 9/12 of the population/years would reject the hypothesis that specialization is due to available environment with $p < 0.05$, with one additional population/year moderately significant with $p = 0.06$. Nestedness does not appear to be driven by the available background to any higher degree than lack of individual identity. Similar to the scenario where individual identity is not important, storks also did not depart from the null distribution for clustering. To obtain a value of clustering < 1 , at least two (or more) niches need to be disconnected from each other. That is, they need to have Jaccard overlap < 0.05 . It is hypothetically possible that individual animals partition environmental space to such a degree that there is very little overlap in their niches, but we don't find that this is true with our populations. Thus, we don't find any evidence for modularity in our study.

Approach:

Fundamentally, I feel the analysis suffers from a lack of null models for determining what would be expected, at random. For example, if use points were randomly assigned to individuals, and the key metrics were re-calculated (e.g., nestedness, etc.), how much 'random' stability would be expected? Re-sampling methods could be extremely helpful in better assessing what is surprising (or not surprising) in these results, for all questions.

Thank you for this suggestion. We followed this advice and performed bootstrapping analyses that tested two different null models. Please see above for further details.

As framed, this study aims to connect individual environmental variation as it scales within “one species”, yet it focuses only on 45 individuals from 3 populations (in one season) of a species that has a very wide range and distribution. Can the results and their interpretation actually scale from individuals to a species, as suggested in the introduction? Or, more accurately, is this analysis only scaling from individuals to a population? And if so, what are we missing from not being able to scale from the population level to the species level?

Thank you for this feedback, we agree it is important not to overstate results. It is true that we do not test populations across this species’ wide geographic extent, but instead examine three separate populations over four years. However, we feel that the stability of the patterns over time and across three populations represent an important result, especially since we are the first to test the geometric relationships among individual Grinnellian niches using hypervolumes and have discovered that the patterns do not match those commonly described in studies of Eltonian niches.

Nevertheless, we have now adjusted the language to better clarify the specific contributions of the study. We call attention to the limitations of the study in terms of number of populations and species, and that further work addressing among-population variation still needs to be conducted in more populations and species in order to understand if the patterns we find common. We feel that our use of increasingly widely available animal movement and remote sensing data make it a very real possibility that this research could be scaled to significantly more populations and species in the future.

Finally, a key aspect of any resource selection analysis is the choice of ‘use’ versus ‘available’ points. While the choice of ‘use’ points in this study are very well justified (based on accelerometer data), I am much more dubious about how ‘available’ points are chosen, particularly as the selection of ‘available’ points can have extremely strong influences on the results of any selection analysis. What perhaps is more worrisome is that the methods never specify how ‘available’ points were chosen. They imply, however, that available points were chosen randomly from all habitats within a certain radius of a centroid of the population. Such an approach would have several problems, however: it assumes that all habitats are equally accessible among individuals and it assumes that all habitats are suitable for use even when clearly not (e.g. if ‘use’ is foraging, then an asphalt parking lot cannot be ‘available’). It seems that for any analysis of ‘individual niches,’ that availability needs to be defined individually, otherwise simple aspects of spatial segregation and home range limitations (i.e., the random distribution of individual on a landscape) can lead to the strong appearance of “specialization” in niches.

We now describe how we sample backgrounds as part of our null model testing (please see methods). In our approach, we do not assume all environments are equally available, or sample from all habitats within a certain radius of the centroid of the population. Instead we define

available environment in terms of the distance an individual might travel from the nest during two-week intervals, informed by each individual stork's movements. Thus, each individual has an available background that is driven by constraints due to breeding phenology (feeding chicks, defending the nest) and the location of its nest in the landscape.

Interpretation:

The only result I don't easily see in the data is the final result, that niches are stable over time. Figure 4 appears to show a huge amount of interannual variation within individuals, as well as temporal trends (e.g. in specialization) at the population level, which further indicates non-stability. The authors will have to do a more convincing job to demonstrate that these niches are stable and consistent, rather than arbitrary. Null models (see above), would help.

We performed null model analysis as suggested and describe our approach and results above.

We feel it is important to point out that this figure shows trends in specialization (as well as nestedness, clustering) over time, not in niche size. Because specialization is calculated relative to the population niche, changes in specialization do not necessarily indicate changes in individual niche size. In this figure, it is more useful to examine changes in rank order, which indicates stability in the relative size of niches among individuals. Individual specialization can vary from year to year, but even though rank order can change somewhat we note that many generalists tend to have low specialization across years (i.e. remain generalists), and specialists tend to have high specialization across years. This is not true for all individuals, as some appear to change strategies, but is true on average, as demonstrated by our repeatability analysis, which quantifies this. We found that specialization has an R of ~50%, which means that 50% of the variance in individual niche specialization is due to consistent differences among individuals. Although this is higher than the average R of 37% found to exist across many taxa (Bell *et al.*, 2009), an R of 50% still means that about half of the variation is due to differences within individuals, which may explain what appears to be variation in the individual trend lines. Finally, we note that at the population level, the level of specialization and nestedness appear stable. This means, for example, that even if individuals tend to change rank order in their level of specialization, at the population level there is similarity in the number and relative sizes of individual niches, even if there is some variation in individual identity. For the specialization metric, there is a slight downward trend over time, but we feel this is likely due to lower sample size in later years. Over time, individuals drop out of the population (they either die or emigrate). If the individual that drops out has the largest niche in the population (and thus drives the size of the population-level niche due to the nested configuration), the overall level of specialization in the population will decrease. We feel that rejection of both null models shows that individual identity is important in driving configuration and that configuration is not completely due to environmental availability, and a repeatability of ~50% shows that consistent, among individual differences in specialization appears to play a strong role in the population-level stability of niche configuration over time.

The application of the results, as presented, to a broad critique of SDMs and how they are used in global change research seems like a stretch and is beyond the bounds of the research findings. I don't disagree that the individual-level behavior may impact SDM performance, but the current analysis sheds limited light on how much of a problem this might actually be, particularly as SDMs are often done at the species level, not the population level (like this study).

We have adjusted language in the discussion to better reflect the particular contributions of the study.

We agree that we do not specifically analyze potential impact to SDMs, but also feel this is outside the scope of the study. We use the assumptions and popularity of SDMs as one factor motivating the study, and hope this will inspire additional research on the integration of individual niches to global change research. However, we do not intend our study to be simply a critique of SDMs, which are an easy target. Instead, we hope our study provides a demonstration of how individual Grinnellian niches might be investigated using increasingly available animal movement and telemetry data, and communicates new ecological insights, in particular our demonstration novel niche configuration patterns through our use of hypervolume analysis and tests of niche configuration.

Additional:

Methods Disclaimer says that code is available from author upon request. As authors often move institutions and may not be easily tracked down, all code should be deposited on a public repository, such as GitHub and linked within the data accessibility page.

We will make all code available on a public GitHub repository

For sampling strategy, it says that storks were haphazardly sampled. Were storks sampled preferentially with respect to age or sex?

Adult storks were trapped using a net gun randomly using a net gun (Super Talon, Zhuhai ZONSO Electronics). It was not possible sample preferentially when trapping since there are no discernable differences between age (above one year) or sex.

References

Araújo, M.S., Bolnick, D.I. & Layman, C.A. (2011) The ecological causes of individual specialisation. *Ecology Letters*, **14**, 948–958.

- Bastille-Rousseau, G. & Wittemyer, G. (2019) Leveraging multidimensional heterogeneity in resource selection to define movement tactics of animals. *Ecology Letters*, **22**, 1417–1427.
- Bell, A.M., Hankison, S.J. & Laskowski, K.L. (2009) The repeatability of behaviour: a meta-analysis. *Animal Behaviour*, **77**, 771–783.
- Beyer, H.L., Haydon, D.T., Morales, J.M., Frair, J.L., Hebblewhite, M., Mitchell, M. & Matthiopoulos, J. (2010) The interpretation of habitat preference metrics under use-availability designs. *Philosophical Transactions of the Royal Society B: Biological Sciences*, **365**, 2245–2254.
- Bolnick, D.I., Svanbäck, R., Fordyce, J.A., Yang, L.H., Davis, J.M., Hulseley, C.D., Forister, M.L. & McPeck, A.E.M.A. (2003) The Ecology of Individuals: Incidence and Implications of Individual Specialization. *The American Naturalist*, **161**, 1–28.
- Colwell, R.K. & Rangel, T.F. (2009) Hutchinson's duality: the once and future niche. *Proceedings of the National Academy of Sciences*, **106**, 19651–19658.
- Costa-Pereira, R., Rudolf, V.H.W., Souza, F.L. & Araújo, M.S. (2018) Drivers of individual niche variation in coexisting species. *Journal of Animal Ecology*, **0**.
- Hutchinson (1957) Concluding Remarks. *Cold Spring Harbor Symposia on Quantitative Biology*, **22**, 415–427.
- Hutchinson, G.E. (1978) An introduction to population ecology.
- Matthiopoulos, J., Fieberg, J., Aarts, G., Barraquand, F. & Kendall, B.E. (2020) Within Reach? Habitat Availability as a Function of Individual Mobility and Spatial Structuring. *The American Naturalist*, **195**, 1009–1026.
- Peterson, A.T., Jorge Soberon, Richard G. Pearson, Robert P. Anderson, Enrique Martinez-Meyer, Miguel Nakamura, & Miguel Bastos Araujo (2011) *Ecological Niches and Geographic Distributions (MPB-49)*, Princeton University Press.
- Soberón, J. (2007) Grinnellian and Eltonian niches and geographic distributions of species. *Ecology Letters*, **10**, 1115–1123.
- Soberón, J. & Nakamura, M. (2009) Niches and distributional areas: concepts, methods, and assumptions. *Proceedings of the National Academy of Sciences*, **106**, 19644–19650.
- Soberón, J. & Peterson, A.T. (2005) Interpretation of models of fundamental ecological niches and species' distributional areas.
- Soberón, J.M. (2010) Niche and area of distribution modeling: a population ecology perspective. *Ecography*, **33**, 159–167.

REVIEWER COMMENTS

Reviewer #2 (Remarks to the Author):

The authors have done a good job in responding to the previous reviews. In terms of substantive changes to the manuscript, I noted the following in particular:

- 1) Stronger theoretical justification outlined in introduction
- 2) More complete methodological information
- 3) Robust evaluation of hypotheses using null model tests
- 4) Re-focused discussion points

Together, I feel the improvements make a much improved manuscript that will be thought provoking for many.

In my reading, I had just two minor line edits:

Line 210: Remove “was”

Line 501: “individual yearly foraging niches during the breeding season” seems like an oxymoron.

Reviewer #3 (Remarks to the Author):

The authors provide an assessment of grinellian niches for storks based on movement data of many individuals over many years. There is much to like in the manuscript, especially as a novel and interesting attempt to better integrate remote sensed data into the ecologies of individual organisms. The results are interesting and should eventually help spur more indepth accounting for how niches are portioned within and between individuals and populations. However there are a number of issues with the current form of the manuscript spanning the analytical, the conceptual, and the organizational. I should also mention for clarity I am a new reviewer for this round.

Analytically the previous round of reviewers were correct to stress the importance of null models. It appears that the authors implemented some null models into their analysis, however as far as I can tell the null models implemented only partially cover the reviewer’s concerns from the last round (concerns that I independently shared when reading the manuscript). In particular, to assess the central claim that individual niches are different null models must be set up to assume a global resource selection function (as is currently done in null model 1) but simultaneously control for the fact that bird home ranges will vary in size, and will not be infinite (as is currently undertaken in null model 2, but not null model 1). Only by accounting for both features can the authors meaningfully test the claim that individuals have different niches, rather than being statistical artifacts of birds occurring in different (and finite) regions of space.

Additionally the methods section was not clear in a number of areas with regards to how the data were treated. For example, why was an interaction between ndvi and tree cover included in the models, but not other interaction terms? How were niche axes whose variables come in totally different units

(distance versus proportion) made to be equivalent? This issue is particularly acute for the distance metrics, as distances should be lognormally distributed, and could on their own induce a skew in apparent niche distributions, which in turn would account for the nested pattern observed in figure 2. Whether this nested pattern is observed across all niche axes or just the distance ones is a relevant question with regards to how generalizable the results are. (Are nested niches common, and applicable to all niche axes, or is it just a statistical artifact when including “distance” axes). Relatedly, whether distance to something is really a proper niche axis in the sense of what Hutchinson envisioned is debatable. I would tend to say “no it’s not a niche axis, with some caveats” but perhaps a cogent mechanistic argument could be made. At the very least better justification for the niche axes chosen is needed. I.e. why is >distance< to urban more appropriate than amount of urban in a given area? And why is >distance< to forest edge more appropriate than amount of forest edge in a given area? In addition to being more coherent as niche axes these flavors of habitat amount would also obviate issues surrounding non-equivalent unit classes. On a related note to the underlying distributions of different niche axes, it would help to show more of the raw data rather than just highly processed model outputs. E.g. what is the distribution of habitat selection actually like in comparison availability? Conceptually more cautious treatment of the niche concept is needed. This is especially relevant in relation to claims that the selected niche axes are important. All models analyzed are correlative, and there is reasons to question whether some (if not most) of the environmental variables used do not directly pertain to a species niche, but instead correlate with the constellation of habitat variables that are actually important. Correlation does not mean that something is an important niche axis. That these are not necessarily strictly speaking niche axes does not sink the central importance of the manuscript on its own, because even if they correlate with niche axes we can make some suppositions about the size and shape of the niche, so long as we clearly state the assumptions underlying those suppositions. But change wording throughout so as not to make overly strong claims that are not directly tested. Finally the manuscript in its current form is often difficult to understand because the methods are at the end. This means that the results presented and discussed are difficult to interpret because the analyzed results are two or three modeling steps removed from the raw data, and the metrics used to evaluate hypotheses are only explained in the methods. To be successful enough methods details need to be provided to the reader in the results to understand the results that are being discussed, and where those results came from.

Summary:

1. Better null models needed, both for “do individuals have different niches” question, and “are niches of individuals consistent between years” question.
2. Better explanation and justification of the 5 niche axes. Current justification is weak leading this reviewer to assume that these environmental variables are being chosen based on data availability rather than the best set of 5 environmental variables that represent known habitat requirements of storks.
3. Better assessment of how individual niche axes drive the results, and whether patterns are due to particular groups of them (distance based axes, versus proportional based measures).
4. Present more from the results of the habitat selection models. How much variation do models explain?
5. Tone down claims that the five remote sensed variables are important niche axes. More carefully think about how unmeasured variables could generate the patterns observed.

Line and section comments for manuscript

Introduction:

~L146: Representing the niche as a hyper volume is not an achievement in and of itself. Focus on why doing so is useful or what you can accomplish with this particular abstraction, rather than the accomplishment of the abstraction itself.

L151: “Can we detect biologically driven...” Comes across as very methods oriented, in that it assumes the variation exists a priori and asks whether you are able to detect the assumed variation.

Epistemologically this approach is scientifically unsound. Further what is “>biologically driven< individual variation” and how do you actually robustly determine whether the variation is “biologically driven”? As opposed to what? Statistical noise and measurement error I guess? Be more clear.

Need to define what you mean by “individual niche” early on. As classical definitions of the niche clearly indicate that only populations have niches.

Results:

L159: Why were these variables selected a priori. Are they measured at the biologically relevant spatial scale? How do you know? ; Table 1 does not exist. These variables need to be described here in the main text, and justification for why they provide a decent representation of all n niche axes needs to be given (at least so that statements about “the niche” as a whole can be made by a sample of 5 variables).

Presenting how much variation in space use is explained by these five variables would help (see below).

L161: Correlation != Causation. “Affirming” way too strong.

L160-170: How do you determine that the associations are driven by the purported niche axis under consideration rather than the distribution of some other unmeasured environmental gradient? If an unmeasured environmental variable is really the cause wouldn't it show up as substantial between individual variations in affiliations on other environmental axes—i.e. the exact pattern you observe. What makes you think that you have captured the 5 key most important environmental variables that are important for habitat selection in storks such that the others' influences are negligible? How much variation in individual foraging location do these five variables explain?

L204-onwards: These results are difficult to understand without having first read the methods. Rewrite to increase clarity.

Needs a null model analysis in order to know what the values of apparent niche specialization would randomly occur if individuals were fully swapping resource selection functions between years. Talk of seven environmental conditions is confusing here, since five were emphasized at the beginning.

Discussion:

L230-235: Raises important concerns regarding correlations between niche axes and causality. The concerns are not addressed in the study.

L250: "These factors do not apply to white storks." No evidence is provided.

Methods:

How much is the nested pattern attributable to generic multidimensionality of the niche (i.e. nestedness coming from all niche axes), or is this a signal being driven by a single niche axis? This speaks to some of the generalities of your findings. Are the niche axis(es) that are causing the nested patterns likely to be causal based on the known biology of these birds.

What is the gps error? How might error interact with the spatial grain of the layers used? The answer will depend on the spatial autocorrelation of the layers (distance metrics less affects than point proportion metrics).

L353: How does partial overlap with another individual ensure access to similar habitat? Explain this more.

How are niche widths along specific axes computed or standardized when niche variables are in unlike units?

Environmental conditions + Resource selection: It's not easy to identify the five purported niche axes evaluated (table 1 did not seem to actually be included in the manuscript). How variables were treated is unclear. What is proximity to forest? How many trees constitute a "forest"? If individuals are inside forests is proximity to forest 0? Or is it negative? There is not enough information here for someone to repeat the analysis. Two of the measures seem to be distances in geographic space, two seem to be proportions, and 1 seems to be a continuous variable that's (presumably) bounded by 0, and some maximum NDVI value. How are niche volumes compared when the units are different? If the niche axes are in different units, and niche widths between individuals across axes are uncorrelated (or worse negatively correlated) wouldn't the overall degree of nestedness be influenced by unit differences?

L412: "We used standardized env data to estimate hypervolumes". What does "standardized" mean here? Are they standardized by subtracting the mean and dividing by the SD? So as to put everything to unit variance and addressing the unit problem mentioned above? Or are they standardized by the availability of the environment in the home range, so as to estimate each species actual niche accounting for environment availability? More detail is needed for the reader to understand the analysis. Also if scaled to unit variance, where does the relevant mean and SD come from? The global distribution of all sites? Within individual populations? The answers to these questions are not obvious,

and decisions could interact by intensifying or reducing the influence of some niche axes over others.

Niche volume: How much did the various niche axes actually contribute to differences in niche volume? Are all axes contributions correlated (suggesting that your selection of only these 5 axes out of the n available doesn't matter) or do different axes tell different stories (suggesting that as selected axes approaches n the story could be quite different)? If trying to make general claims about the niche per se, rather than particular purported niche axes that are chosen based on data availability then addressing this pitfall is crucial.

L426-430: If calculating specialization based on all individuals in a population, won't population niche width be a function of number of individuals included, such that individuals with populations where more individuals were sampled will appear to be more specialized than more poorly sampled populations?

Null model 1: Does this null model effectively test the hypothesis? By sampling from all samples in a population the movement/home range constraints are eliminated, meaning you will be sampling from a larger potential environment than any individual will have access to. To actually test an individual effect you would need to constrain movement distance while also applying a global model for resource/env selection. For example, a parametric version of this null model would take a population-wide selection function, pick a random point in the landscape, select a movement distance/home range size/shape from the empirically observed individuals, and then probabilistical sample the area using the global selection function to derive a null distribution of selection. In this case the range-size/movement is what was observed in the data, but the selection function is global rather than individual. Now is the observed distribution of individual specialization/nestedness different from the null distribution? (I.e. a good test of the hypotheses that individuals are truly different in their niches from random variation derived from a global resource selection function needs to account for both properties of the current null model 1 and null model 2).

Repeatability: Some explanation of the intraclass correlation coefficient needs to be given in the results, so that the reader can understand what that results means without having to read the methods all the way to the end, and then return to the results. If the methods are placed at the end then enough method detail (about this, but also other features, like which niche axes were examined, etc) need to be provided in the results for the reader to understand generally where the results come from, and what various quantities actually mean.

Figures:

Figures do not show enough data for the reader to be confident in results. Figure 2 should include null distributions. Figure 3: What is the null distribution of repeatability if individuals yearly specialization was sampled from alternative individual's resource selection functions (but constraining home range size?)

In the supplement we learn that there was an "interaction term for percent forest and NDVI". Why is

this not in the results, and why was this included?

Supplemental figures appear to be multiple figures for every caption, making it very difficult to identify which figures are "S1" and "S2".

S3. P can not equal 0.

S4. Provide best fit line, and significance of the relationship. "diver" -> "driver".

Table S1: My supposition is that this is actually supposed to be "Table 1" referred to in text? RE: Bare ground. My understanding of the Hansen 2013 dataset was that it included tree cover, and non-tree cover, not bare ground per se. Is this not correct? In the maintext the rationale behind these 'niche axes' could be better described. Is "Distance to X" an actual niche axis in the sense of Hutchinson etc? These variables feel like widely available globally sensed data that presumably correlate with the actual niche axes that this species is responding to, but that do not actually in and of themselves represent true niche axes. Justification for use of distance as a niche axis in particular needs to be justified.

Response to last round of review:

As you state in your response to reviewers applying the hutchinsonian niche concept to individuals runs into problems in so far as individuals do not have population growth rates. However, the closest thing to population level growth at the individual is fitness. You are presumably estimating the relationship of fitness proxies (with the associated assumptions therein). If your "individual niches" do not track individual absolute fitness then they are fairly useless as niche measurements.

"Thus, we do not simply consider any environmental conditions the individual experiences as part of its niche, but only those conditions that allow the individual to survive and reproduce. This is one reason why we focus on foraging locations during the breeding period in our study, and have filtered out other locations such as sitting, preening, flying, etc. By focusing on the environmental characteristics of an individual's foraging locations, which are closely tied to the individual's ability to survive and reproduce, we estimate individual realized niches."

These statements are fine but they are prone to miss key components of the environment that will affect survival and reproduction other than just foraging.

"Based on these definitions, it is easy to see that our study, which examines environmental axes (such as tree cover, or distance to urban) that are not changed by the presence of individuals (at the time scale of our study) estimate individual Grinnellian niches." More nuance in paper is needed to acknowledge that the grinnellian niche variables measured are correlative, and any apparent grinnellian niche shifts could in fact be due to other environmental variables, or to the eltonian niche variables as originally suggested by reviewer two. IF these variables represent niche axes for storks I certainly agree that they belong in the Grinnellian category. However, not every scenopoetic variable that correlates with presense/survival/reproduction is a relevant niche axis. To give a stupid made up example, the local abundance of Samsung brand cell phones is (presumably) not a real niche axis for these storks (5G conspiracies notwithstanding), even though stork occurrence, abundance, survival, and reproduction

almost certainly all correlate with this variable. This undermines your claims that your land-cover variables are definitively relevant niche axes for these storks, as (like with cell phones) the apparent specialization on them (or away from them) could be due their correlation with other true niche variables.

Reviewer #4 (Remarks to the Author):

I have read with great interest the manuscript titled "Individual environmental niches in 1 mobile organisms" by Carlson et al. et al.

I did notice this was a revised version, and I had access to all previous review material, including Reviewers' comments, Authors' rebuttal, and revised manuscript.

I have some more comments that should be addressed to cement the clarity of the paper:

L63-64 this is confusing because you do not in fact address the connection between Eltonian (trophic) and Grinnellian (geophysical) niches later on, but model individual Grinnellian niches and discuss their relative position within the population-level Grinnellian niche.

L389 The Discussion ends abruptly. I would add a take-home message paragraph based on your data and results, rather than ending on recommendations for future research.

L429 I don't understand why you even need to define a home range if you perform a step-selection function? The domain of availability is defined by the step-length distribution, and there is no need to build a home range; but I might have missed something.

L 443 SSFs are not themselves niche models, but...

L444 do you have a reference to back up this approach of using a SSF to identify niche axes?

The methods in the Environmental conditions and Resource selection sections are not very clear. Maybe swapping these 2 sections around could help. First you present the RS analysis, and then you introduce the environmental variables (the term conditions is a bit odd in this context) at what is now L495, between the amt packages and distance to nest.

485 ... we performed a resource selection analysis. Resource selection analyses seek to ...

L505 intervals did not overlap 0

L517 we used to calculate

Reviewer #2 (Remarks to the Author):

The authors have done a good job in responding to the previous reviews. In terms of substantive changes to the manuscript, I noted the following in particular:

- 1) Stronger theoretical justification outlined in introduction
- 2) More complete methodological information
- 3) Robust evaluation of hypotheses using null model tests
- 4) Re-focused discussion points

Together, I feel the improvements make a much improved manuscript that will be thought provoking for many.

We thank the reviewer for their insightful comments on the first review and for their supportive comments for this latest version.

In my reading, I had just two minor line edits:

Line 210: Remove “was”

Thank you, we have made this update.

Line 501: “individual yearly foraging niches during the breeding season” seems like an oxymoron.

We removed “yearly” and updated this to “individual foraging niches during the breeding season”

Reviewer #3 (Remarks to the Author):

The authors provide an assessment of grinellian niches for storks based on movement data of many individuals over many years. There is much to like in the manuscript, especially as a novel and interesting attempt to better integrate remote sensed data into the ecologies of individual organisms. The results are interesting and should eventually help spur more indepth accounting for how niches are portioned within and between individuals and populations. However there are a number of issues with the current form of the manuscript spanning the analytical, the conceptual, and the organizational. I should also mention for clarity I am a new reviewer for this round.

R1) Analytically the previous round of reviewers were correct to stress the importance of null models. It appears that the authors implemented some null models into their analysis, however as far as I can tell the null models implemented only partially cover the reviewer’s concerns from the last round (concerns that I independently shared when reading the manuscript). In particular, to assess the central claim that individual niches are different null models must be

set up to assume a global resource selection function (as is currently done in null model 1) but simultaneously control for the fact that bird home ranges will vary in size, and will not be infinite (as is currently undertaken in null model 2, but not null model 1). Only by accounting for both features can the authors meaningfully test the claim that individuals have different niches, rather than being statistical artifacts of birds occurring in different (and finite) regions of space.

We thank the reviewer for this suggestion. We agree that using a population-level RSF provides additional realism to the null model and will help tease apart the effects of environmental availability from individual selection when examining the relationships among individual usage distributions. We implemented a third null model by interpreting resource selection through weighted distribution theory (Johnson et al., 2008).

$$f_u \sim w * f_a$$

Where f_u is the distribution of habitats at used locations, f_a is the distribution at available locations, and w is the resource selection function. We updated the methods to include the following description.

“Finally, we combined aspects of the first two null models in order to test a third null model. Specifically, we used a population-level resource selection function, as in the first null model, but constrained the available distribution using the buffers described in the second null model. We implemented the third null model by interpreting resource selection according to weighted distribution theory (see section Resource Selection, above). In a typical resource selection analysis, we use f_u and f_a to estimate w . For the null model, we take as w the back-transformed mean of the selection coefficients for the individuals in each population/year. We then use this to sample from f_a in order to generate f_u . By sampling from the available backgrounds at each nest site according to a population-level RSF, we produce usage distributions for each site as though they are generated by the same individual (in this case, the average stork). We then used these distributions to construct hypervolumes as described above.”

This null model fits the criteria described by the reviewer, as it uses a global resource selection function (as in the first null model), but constrains the available distribution according to the storks' home range (as in the second null model).

We have included the distribution of configuration metrics under the third null model in Fig. S3. We find that, similar to the second null model, under the third null model the null hypothesis is rejected in ¾ of population/years for the specialization metric. For the nestedness metric, the null hypothesis is rejected in 11/12 population/years, similar the null hypothesis for both the first and second null models, which is rejected 12/12 times. Finally, like the second and third null models, the null hypothesis is never rejected for the clustering metric.

Thus, we find that even if we moved the average stork to each location and constrained its movements according to breeding phenology and other factors, this alone would not explain the observed values of specialization and nestedness. Instead, there is evidence that individual

differences in use of environments has a statistically significant contribution to environmental niche specialization and configuration.

R2) Additionally the methods section was not clear in a number of areas with regards to how the data were treated. For example, why was an interaction between ndvi and tree cover included in the models, but not other interaction terms?

We based our covariates and interaction terms on our knowledge of white stork biology. In particular, we hypothesized that storks would be attracted to grassland patches with relatively high NDVI, but that this signal could be confused with forested pixels with similar NDVI. We attempted to account for this issue by including a term that adjusts for an interaction between NDVI and percent of tree cover. We hypothesized that this would result in a negative interaction that would weaken responses to pixels with high NDVI if accompanied by high tree cover. We note that in most cases where the interaction term is statistically significant, it is negative, supporting our hypothesis (Fig S1). We did not include other interactions because we did not hypothesize that any other potential interactions were important, and we wanted to avoid unnecessarily increasing model complexity by adding extraneous interaction terms.

We have updated the methods section to reflect this explanation:

“Additionally, we hypothesized that storks are attracted to patches of grassland with relatively high NDVI, but that this would not similarly be the case for forested patches with high NDVI. Thus, we should see a weaker response when a pixel with moderate to high NDVI also contains high percent tree cover. To account for this effect, we included a term that adjusts for interaction between NDVI and percent of tree cover.”

R3) How were niche axes whose variables come in totally different units (distance versus proportion) made to be equivalent?

Thank you for this question. We now clarify this through the following addition to the methods section:

“We standardized (i.e. z-transformed) all variables to have mean 0 and standard deviation 1. This results in units of standard deviations for all variables. This is the recommended procedure for linear models (Schielzeth, 2010) as well for hypervolume estimation (Blonder et al., 2018). We standardized over the full dataset, in order to maintain comparability among individuals and sites.”

R4) This issue is particularly acute for the distance metrics, as distances should be lognormally distributed, and could on their own induce a skew in apparent niche distributions, which in turn would account for the nested pattern observed in figure 2. Whether this nested pattern is observed across all niche axes or just the distance ones is a relevant question with regards to

how generalizable the results are. (Are nested niches common, and applicable to all niche axes, or is it just a statistical artifact when including “distance” axes).

Thank you for bringing up this important observation. It is useful to investigate whether the nested pattern is caused by the distance only variables. We performed the full niche analysis using only the distance variables (distance to forest, distance to urban), and only the non-distance variables (percent tree cover, percent bare ground, NDVI). The figure below summarizes the results.

Based on these results, it does not appear that higher specialization or nestedness is driven by the distance variables.

R5) Relatedly, whether distance to something is really a proper niche axis in the sense of what Hutchinson envisioned is debatable. I would tend to say “no it’s not a niche axis, with some caveats” but perhaps a cogent mechanistic argument could be made. At the very least better justification for the niche axes chosen is needed. I.e. why is >distance< to urban more appropriate than amount of urban in a given area? And why is >distance< to forest edge more appropriate than amount of forest edge in a given area? In addition to being more coherent as niche axes these flavors of habitat amount would also obviate issues surrounding non-equivalent unit classes.

We thank the reviewer for this interesting comment. Distance variables have different characteristics than density or discrete landcover variables and it is useful to discuss these

differences. For distance variables, the researcher often does not believe that the animal is interested in the object at zero distance, but rather believes the animal interacts with some resource or risk gradient that extends from that object. For example, “distance to road” is a common variable in resource selection studies (e.g. Nielson & Sawyer, 2013). The road itself is often not the subject of the animal’s interest, but rather the variable of interest is a risk gradient that extends from the road, generating a landscape of fear (Hertel et al., 2019). It is generally hard to directly measure this risk gradient, but relatively easy to measure roads and create distance variables. If there is sufficient belief that the risk gradient correlates with the distance to the road, distance to road can be used as a measurement of the amount of risk at each spatial location.

Distance variables are very common in resource selection studies (Thurfjell et al., 2014). Examples include distance to water (Duduś et al., 2014), distance to agriculture (Carpenter et al., 2010), distance to natural areas (Clark et al., 2015), as well as the two variables that we employ, distance to forest (Chetkiewicz & Boyce, 2009; Clark et al., 2015) and distance to urban (Carpenter et al., 2010; Duduś et al., 2014; Rainho & Palmeirim, 2011). It is more difficult to find examples using distance variables in hypervolumes, since we are among the first to estimate niches using the Grinnellian variables commonly used in resource selection studies. However, we can look to Hutchinson to find a good example, as well as potentially answer the Reviewer’s question regarding what Hutchinson would have thought about distance variables.

In Hutchinson’s most lengthy treatment of the niche (Hutchinson, 1978), he is largely silent regarding distance variables as niche axes. However, he does mention one particular distance variable as a niche axis: distance to the ground, or height (p. 187). Although often not thought about as a distance variable, height shares many properties with other distance variables. Like other distance variables, the animal is not usually interested at the object at zero distance (in this case, the ground). And like other distance variables, we use distance to as a proxy for a gradient of resources, risks, or competitive intensity. Thus, although we don’t have a definitive answer, Hutchinson’s use of height implies that he may have considered other distance variables as legitimate niche axes.

Although we have shown that many researchers, possibly including Hutchinson, have used distance variables, this by itself is not sufficient to justify their use in our study. However, we feel that many of the arguments above apply to White Storks and specifically to our study. The resource and risk gradients extending from urban and forest are important to stork foraging behavior and habitat selection. The gradient of resources and risks extending from forests include predation risk, ecotone prey resources, and thermal gradients. Similarly, the gradient extending from urban include anthropogenic subsidies such as prey resources in meadows, or water availability (e.g. irrigation ditches). Additionally, these axes should differentiate individual habitat and resource use. For example, storks may differ in how they tradeoff resource acquisition with predation risk, which could influence the distance at which they forage near a forest edge.

Finally, the reviewer asks about the relative importance of density variables vs. distance variables. We believe that both variable types can be useful. Density variables directly capture attraction to, or repulsion from, the variable in question, whereas distance variables capture characteristics related to the spatial arrangement of resources. Both types of variables can be important in models. Clark et al. (2015) included both distance and density variables for six habitat types, and found the best performing models included the distance variable, but not the density variable, for each habitat. Rainho & Palmerimim (2011) found that both landcover and distance variables were important, and that models containing distance variables performing significantly better than those without. In our study, we include both distance to forest and percent forest cover (a variable similar to density of forest). We find that both variables are generally significant in our SSF models, lending support to the hypothesis that these variables capture different but important habitat characteristics, and are both relevant as niche axes.

R6) On a related note to the underlying distributions of different niche axes, it would help to show more of the raw data rather than just highly processed model outputs. E.g. what is the distribution of habitat selection actually like in comparison availability?

We include distributions of all untransformed habitat variables, for each individual/year (Fig. S6). This figure allows one to see the shape of the underlying univariate habitat distributions as well as differences among the individuals in the same population.

R7) Conceptually more cautious treatment of the niche concept is needed. This is especially relevant in relation to claims that the selected niche axes are important. All models analyzed are correlative, and there is reasons to question whether some (if not most) of the environmental variables used do not directly pertain to a species niche, but instead correlate with the constellation of habitat variables that are actually important. Correlation does not mean that something is an important niche axis. That these are not necessarily strictly speaking niche axes does not sink the central importance of the manuscript on its own, because even if they correlate with niche axes we can make some suppositions about the size and shape of the niche, so long as we clearly state the assumptions underlying those suppositions. But change wording throughout so as not to make overly strong claims that are not directly tested.

Thank you for bringing up this important point. It is certainly true that through an observational study and a correlation model we can't conclude that there is a causal relationship between these variables and the intensity of habitat use. As with all studies of niches, we selected variables using our best judgement based on the biology of our study organism and the current capacity to measure the environment. Like a hypothesis test, we can't directly confirm that our choices were correct, but we can reject variables that are not correlated with intensity of use. Thus, results showing statistically significant selection supports the notion that we chose our variables correctly.

We have softened the language, and removed words such as "affirming", and "validating", and altered how we describe our interpretation of significant selection in the SSF model.

For example, we now include the following in the methods section: “We interpreted statistically significant selection for or against a given environmental variable as evidence that the variable is, or is strongly correlated with, a niche axes for that individual.”

R8) Finally the manuscript in its current form is often difficult to understand because the methods are at the end. This means that the results presented and discussed are difficult to interpret because the analyzed results are two or three modeling steps removed from the raw data, and the metrics used to evaluate hypotheses are only explained in the methods. To be successful enough methods details need to be provided to the reader in the results to understand the results that are being discussed, and where those results came from.

We addressed all specific points of confusion (see below), and added additional context where we imagined a reader might be confused. We are happy to make more improvements if the reviewer has additional specific comments.

Summary:

R9) 1. Better null models needed, both for “do individuals have different niches” question, and “are niches of individuals consistent between years” question.

Please see the answer to “R1”, above, where we discussion details of “do individual have different niches”.

In addition, we performed repeatability analysis under each of the three null models, and show the results in Fig. S7. In all cases, the observed value of repeatability rejects the null hypothesis ($p < .01$). Depending on the nature of the null model, observed repeatability is either significantly above, or significantly below the null distribution.

Recall that repeatability is among-individual variance / (among-individual variance + within-individual variance). Thus, decreases in among-individual variance (relative to within-individual variance) will result in lower repeatability, whereas decreases in within-individual variance (relative to among-individual variance) will result in higher repeatability.

In the first null model (individual identity), we have randomized the individual identities. This will in turn reduce among-individual variance, resulting in a null distribution of repeatability that is lower than the observed repeatability.

In the second null model (environmental availability), we select the used distribution from buffers around the nest in proportion to environmental availability. Although this should reduce among-individual variance, it should reduce within-individual variance to a greater degree, because the land cover layers are mostly static (there are greater differences among sites than within sites over time). Thus, we would expect to see higher repeatability under the null model compared to observed value.

In the third null model (common SSF), we define available environment according to buffers around the nest (as in the second null model), but we use a population-level SSF to select the used distribution. Like the second null model, we would expect this to result in reduced within-individual variation (compared to among-individual variation). This is because we are using mean values to define the common SSF, and central tendency will reduce the variation across years.

R10) 2. Better explanation and justification of the 5 niche axes. Current justification is weak leading this reviewer to assume that these environmental variables are being chosen based on data availability rather than the best set of 5 environmental variables that represent known habitat requirements of storks.

The selection of niche axes is a careful balance between our knowledge of the biology of the species and the capability to measure the environment. So, while data availability was certainly a factor, we also carefully considered stork biology and scale.

For example, we considered but rejected climate variables (temperature, precipitation) as these variables were too coarse to offer ecologically meaningful information at the scale of patch use within individual home ranges. Generally, we recognized coarser-grain (250m) variables as too insufficiently detailed and spatially autocorrelated to be statistically interpretable at the scale of individual foraging movements. We found 30m resolution characterizations of carefully selected environmental attributes to offer the currently best-possible combination of spatial detail reflecting stork patch size choice and data availability. In order to capture relevant factors as appropriately as currently possible, we did not simply accept remote sensing products that are generally available, but instead created our own layers when our desired layer did not exist. Three out of the five layers in our analysis were custom developed for this research: distance to urban, distance to forest, and fused Landsat 7 and 8 in order to have a denser NDVI time-series than is currently available.

We also now include maps of all environmental variables, for each of the three populations (Fig. S8, Fig. S9, Fig. S10). These maps highlight the heterogeneity in the landscapes, which provide storks with many opportunities to specialize on various microhabitats.

R11) 3. Better assessment of how individual niche axes drive the results, and whether patterns are due to particular groups of them (distance based axes, versus proportional based measures).

We ran distance and non-distance variables separately and did not find that distance variables drive the results. For additional information, please see our answer to (R4) above.

R12) 4. Present more from the results of the habitat selection models. How much variation do models explain?

The use resource selection analysis for the purpose of confirming the relevance of the expert-selected set of predictors. We present full details on the parameter values, confidence intervals and statistical significance of each variable – individual – year combination in Figure S1. Goodness of fit (r^2 style) characterizations for resource selection analysis have limited value for interpretation due to issue of data non-independence. Across individuals, each of the variables shows a large number of statistically strongly significant associations. We would be happy to add a particular goodness of fit metric as yet further support to the supplement, but given their limitations would welcome guidance from the reviewer as to which particular metric they would like to see.

R13) 5. Tone down claims that the five remote sensed variables are important niche axes. More carefully think about how unmeasured variables could generate the patterns observed.

Please see our answers to R7, above, and to L161 and L160-170, below.

Line and section comments for manuscript

Introduction:

~L146: Representing the niche as a hyper volume is not an achievement in and of itself. Focus on why doing so is useful or what you can accomplish with this particular abstraction, rather than the accomplishment of the abstraction itself.

We have updated this text so that it no longer stresses the hypervolume, but rather on why it is a useful framework.

“Although there have been important advances in using these types of data to estimate individual variation in behavior, including resource selection and repeatability, no study has represented individual niches in a multidimensional framework that allows examination of the geometric configuration among individual niches”

L151: “Can we detect biologically driven...”. Comes across as very methods oriented, in that it assumes the variation exists a priori and asks whether you are able to detect the assumed variation. Epistemologically this approach is scientifically unsound. Further what is “>biologically driven< individual variation” and how do you actually robustly determine whether the variation is “biologically driven”? As opposed to what? Statistical noise and measurement error I guess? Be more clear.

Thank you for this comment. We agree that “biologically driven...” is confusing. We have simplified the question to the following:

“(1) Are individual Grinnellian niches more specialized than expected by chance?”

R14) Need to define what you mean by “individual niche” early on. As classical definitions of the niche clearly indicate that only populations have niches.

We have included the following definition early in the introduction.

“The concept of an individual niche is still in need of formal characterization. Here, we employ a working definition: an individual Grinnellian niche is the set of all points in environmental (niche) space, as defined by Grinnellian axes, that permit an individual to survive and reproduce.”

Results:

L159: Why were these variables selected a priori. Are they measured at the biologically relevant spatial scale? How do you know? ;

We have updated this paragraph to provide more information to the reader regarding the habitat variables. We now include the following in the first paragraph of the results:

“We linked these locations to carefully selected, remotely sensed environmental variables that represent known habitat associations for the species. These variables, all captured at 30m resolution, address key aspects of foraging habitat structure, quality, and access (16-day NDVI, percent of tree and bare ground cover, distance to built-up areas and forest; Figure S6, Table S1).”

R15) Table 1 does not exist. These variables need to be described here in the main text, and justification for why they provide a decent representation of all n niche axes needs to be given (at least so that statements about “the niche” as a whole can be made by a sample of 5 variables). Presenting how much variation in space use is explained by these five variables would help (see below).

We apologize for this mistake. The text should have referred to Table S1. We have updated this throughout the text. We now list the variables in the results section and refer readers to the methods section where we provide further justification of the variables (please see answer to L159 for further details).

L161: Correlation != Causation. “Affirming” way too strong.

We have removed this language. Please see response to R7, above, for additional discussion of this theme.

L160-170: How do you determine that the associations are driven by the purported niche axis under consideration rather than the distribution of some other unmeasured environmental gradient? If an unmeasured environmental variable is really the cause wouldn't it show up as substantial between individual variations in affiliations on other environmental axes—i.e. the exact pattern you observe. What makes you think that you have captured the 5 key most important environmental variables that are important for habitat selection in storks such that

the others' influences are negligible? How much variation in individual foraging location do these five variables explain?

We chose niche axes based on our knowledge of stork ecology and behavior, and given the current capacity to measure the environment. There are certainly some variables that we could imagine might improve our models. For example, we know that storks are strongly attracted to recently cut meadows, and that they tend to forage for worms after rain in damp, open habitats. However, the technology does not yet exist to capture these specific, dynamic habitats, and furthermore we believe that many of these habitat factors will be captured by the variables we chose, such as percent forest and percent of bare ground.

One could always imagine an unmeasured variable that might drive habitat use, however we do not believe the existence of such a variable would change our findings. If an unmeasured variable strongly drove habitat selection, but was uncorrelated with our habitat variables, we should not see an association between our variables and habitat selection. On the other hand, if the unmeasured variable was correlated with our habitat variables, our variables should capture the effects of this unmeasured variable and the overall pattern in our results should be similar.

We would also like to bring up the point that the goal of niche analysis is not necessarily to find the key most important set of axes, but instead to use axes that will provide some mechanism to understand the response that is the subject of the study. Hutchinson (1978) points this out in his discussion of niche dimensionality: "although to define the tolerances and needs of a single species completely would indeed require a very large value for n , the study of the difference between two species can usually be conducted in a niche space of two or three dimensions". Here, the goal of our study is to use niche analysis to understand whether there is individual differential use of habitats during the foraging process. In line with Hutchinson's philosophy, we chose variables that we felt foraging storks would respond to, and that would have some capacity for individual differentiation.

L204-onwards: These results are difficult to understand without having first read the methods. Rewrite to increase clarity.

We responded to all specific comments and added additional clarifications where it seemed appropriate.

R16) Needs a null model analysis in order to know what the values of apparent niche specialization would randomly occur if individuals were fully swapping resource selection functions between years.

Please see our response to (R1) for discussion regarding the suggested null model.

R17) Talk of seven environmental conditions is confusing here, since five were emphasized at the beginning.

We have provided additional clarity when discussing these variables.

“We then used the five main environmental variables (and not the two additional variables, distance to the nest and NDVI percent tree cover interaction) to construct and compare individual hypervolumes.

In addition, we added the following to the methods section:

“The addition of these two additional terms (distance to nest and NDVI/tree cover interaction) resulted in a total of seven terms in the SSF model, although we only use the five main terms as niche axes in our estimation of hypervolumes”

Discussion:

L230-235: Raises important concerns regarding correlations between niche axes and causality. The concerns are not addressed in the study.

It was not our intention for this paragraph to address issues of causality. As the reviewer correctly remarks elsewhere, we can't infer causality in an observational study. Rather, the intention of this paragraph was to highlight the importance of performing both resource selection analysis, as well as hypervolume-based analysis when attempting to understand individual habitat requirements. While we believe that evaluating our choice of niche axes using resource selection is a strength of our study, we understand that these methods are still correlational and do not make claims about causality. To this end, we have softened our language regarding our inference of statistically significant selection. Please see responses to R7, L161 and L160-170 for further details.

L250: “These factors do not apply to white storks.” No evidence is provided.

We have updated the discussion to provide additional evidence for our conjecture.

“In some systems, social dominance hierarchy or the existence of distinct morphotypes causes consistent niche structure(Bolnick et al., 2003). Breeding white storks often forage alone but are also known to form aggregations(Carrascal et al., 1990). Although conditions exist for social dominance to occur while aggregating(Piper, 1997), we are unaware of any evidence for social dominance in white stork foraging. Furthermore, dominance effects, if they occur, should affect the use of resources within a patch (e.g. feeding rates on an animal carcass(Marzlufi & Heinrich, 1991; van Overveld et al., 2018)), whereas variation in environmental niches is due to differential use among patches. Thus, it is unlikely that dominance hierarchies play a significant role in environmental niche specialization. Likewise, we are unaware of the existence of distinct morphotypes (e.g., sympatric morphotypes in populations of lake trout(Moore & Bronte, 2001)), thus it is unlikely that morphology is an important factor.”

Methods:

R18) How much is the nested pattern attributable to generic multidimensionality of the niche (i.e. nestedness coming from all niche axes), or is this a signal being driven by a single niche axis? This speaks to some of the generalities of your findings. Are the niche axis(es) that are causing the nested patterns likely to be causal based on the known biology of these birds.

We agree with the reviewer that this is an interesting and important question. We have provided some analysis of distance vs. non-distance variables (please see R4 for more information), but we feel that providing a through answer to this question is outside the scope of this study. In our study, we focus on describing a pattern that is at odds with the conventional assumptions of how individuals partition niche space. Although we discuss several possible mechanisms for these patterns, we don't directly test them in the study. We feel these are excellent topics for future study.

R19) What is the gps error? How might error interact with the spatial grain of the layers used? The answer will depend on the spatial autocorrelation of the layers (distance metrics less affects than point proportion metrics).

Thank you for this question, we now include this information in the methods:

"GPS locations had a 50th percentile spatial accuracy of <3.6m (50% of the points are within 3.6 m of the true location), and 95th percentile accuracy of <19 m."

Given that the linear accuracy of our GPS locations is smaller than the resolution of our environmental variables (30m), we feel that error should have negligible impact. In addition, although there are many sharp boundaries in our landscapes, many of the habitat features are larger than 30m. Thus, any errors that extend beyond the boundaries of the correct pixel have a high likelihood of landing in a pixel with similar environmental characteristics. Finally, we removed extreme outliers (locations that resulted in unreasonable speed), which further reduces the chance that error will result in an environmental value significantly different than the conditions at the true location.

L353: How does partial overlap with another individual ensure access to similar habitat? Explain this more.

We used the partial overlap of home range as a simple indication that individuals were in the same population (e.g. interacted, shared similar resources). For the question of access to resources, we really want to know *could* an individual stork access the habitat, even if, based on the empirical data, it chose not to. Based on stork's immense movement capacity we might consider resources far outside their home range as accessible. However, as a conservative measure, and to recognize other constraints (e.g. nest defense) we use the rule that the stork's home range should have some overlap with another storks.

R20) How are niche widths along specific axes computed or standardized when niche variables are in unlike units?

We performed z-transformation of all variables. Please see R3 for further details.

R21) Environmental conditions + Resource selection: It's not easy to identify the five purported niche axes evaluated (table 1 did not seem to actually be included in the manuscript). How variables were treated is unclear. What is proximity to forest? How many trees constitute a "forest"? If individuals are inside forests is proximity to forest 0? Or is it negative? There is not enough information here for someone to repeat the analysis. Two of the measures seem to be distances in geographic space, two seem to be proportions, and 1 seems to be a continuous variable that's (presumably) bounded by 0, and some maximum NDVI value. How are niche volumes compared when the units are different? If the niche axes are in different units, and niche widths between individuals across axes are uncorrelated (or worse negatively correlated) wouldn't the overall degree of nestedness be influenced by unit differences?

We apologize that this information was not readily accessible. Table S1 (which was mis-labeled Table 1) contains this information. We standardized(z-transformed) all variables (please see R3 for further details).

L412: "We used standardized env data to estimate hypervolumes". What does "standardized" mean here? Are they standardized by subtracting the mean and dividing by the SD? So as to put everything to unit variance and addressing the unit problem mentioned above? Or are they standardized by the availability of the environment in the home range, so as to estimate each species actual niche accounting for environment availability? More detail is needed for the reader to understand the analysis. Also if scaled to unit variance, where does the relevant mean and SD come from? The global distribution of all sites? Within individual populations? The answers to these questions are not obvious, and decisions could interact by intensifying or reducing the influence of some niche axes over others.

We z-transformed each environmental variable, over the full dataset. Please see R3 for additional details.

R22) Niche volume: How much did the various niche axes actually contribute to differences in niche volume? Are all axes contributions correlated (suggesting that your selection of only these 5 axes out of the n available doesn't matter) or do different axes tell different stories (suggesting that as selected axes approaches n the story could be quite different)? If trying to make general claims about the niche per se, rather than particular purported niche axes that are chosen based on data availability then addressing this pitfall is crucial.

Please see our answer to R18 regarding investigation of different niche axes.

L426-430: If calculating specialization based on all individuals in a population, won't population niche width be a function of number of individuals included, such that individuals with

populations where more individuals were sampled will appear to be more specialized than more poorly sampled populations?

It is true that sampling more individuals could increase the size of the population niche, but this depends on how the niches are configured. Our finding that niches are nested reduces the chance that additional sampling will always increase the size of the population niche. For example, if we add an individual that happens to be a generalist (relative to the other individuals), then the size of the population niche should increase. But we see that most individuals are nested within the niche of the largest individual, and sampling one of these more relatively specialized individuals won't increase the population niche as much. In our empirical results, we do see this pattern. Comparing the same population across years, although patterns are remarkably stable, we do see a decline in overall specialization (Fig. 3a), which may occur because generalists are randomly dropped from the population over time. However, despite the fact that specialization has some dependence on the number of individuals, we note that nestedness is even more stable over time (Fig. 3b). Thus, we see the nested configuration regardless of the number of individuals sampled.

Null model 1: Does this null model effectively test the hypothesis? By sampling from all samples in a population the movement/home range constraints are eliminated, meaning you will be sampling from a larger potential environment than any individual will have access to. To actually test an individual effect you would need to constrain movement distance while also applying a global model for resource/env selection. For example, a parametric version of this null model would take a population-wide selection function, pick a random point in the landscape, select a movement distance/home range size/shape from the empirically observed individuals, and then probabilistically sample the area using the global selection function to derive a null distribution of selection. In this case the range-size/movement is what was observed in the data, but the selection function is global rather than individual. Now is the observed distribution of individual specialization/nestedness different from the null distribution? (I.e. a good test of the hypotheses that individuals are truly different in their niches from random variation derived from a global resource selection function needs to account for both properties of the current null model 1 and null model 2).

Thank you for this excellent suggestion. As suggested, we implemented a null model that accounts for the properties of both null model 1 and null model 2. Please see R1 for the full details.

Although we adopted the excellent ideas of using a population-level RSF while constraining potential movements, we did not adopt some of the additional suggestions described here because we did not feel these steps were appropriate for the null model. Specifically, the reviewer suggested above that we also 1) pick available distributions from random points in the landscape, and 2) allow buffers sizes to randomly vary.

First, our study is focused on individual variation in selection of habitat patches within existing home ranges, not the selection of home ranges within the larger landscape. We felt that picking

random locations in the landscape would obfuscate interpretation because the null model would incorporate elements of home-range level selection. For example, storks do not nest in forests. Thus, a location randomly placed within a forested area would provide information about home-range level selection, but would add noise to the model that would limit its ability to detect individual differences in selection within home ranges.

Second, our use of spatiotemporal buffers to delineate available habitat was specifically to partition out constraints that are not strictly due to resource selection. For example, storks might need to stay closer to the nest due to the age of their chicks, to guard against hostile takeover from another stork, or to limit nest predation. These constraints have more to do with breeding phenology and the density of competitors and predators than with foraging patch selection. These factors should be largely independent of foraging habitat selection *per se*, although we can imagine some interdependence. For example, early in the breeding period storks might select certain habitats because chicks have restricted diets (e.g., storks might select more bare ground to forage for worms because of chicks' restricted gape size). For the null model, we make the simplifying assumption that these constraints are independent of habitat selection, because we want to focus on variation in foraging preferences and not these other constraints.

Repeatability: Some explanation of the intraclass correlation coefficient needs to be given in the results, so that the reader can understand what that results means without having to read the methods all the way to the end, and then return to the results. If the methods are placed at the end then enough method detail (about this, but also other features, like which niche axes were examined, etc) need to be provided in the results for the reader to understand generally where the results come from, and what various quantities actually mean.

We have made several updates to the results to improve readability, including this suggestion regarding repeatability. Specifically, in response to this suggestion, we updated the methods:

“After observing similar resource selection and niche configuration patterns among all populations and years (Fig. S2), we conducted a formal analysis of the consistency of these patterns over time using the Intraclass Correlation Coefficient (Nakagawa & Schielzeth, 2010). The ICC is a common metric of repeatability and measures the proportion of population variance explained by variance among individuals.”

Figures:

Figures do not show enough data for the reader to be confident in results. Figure 2 should include null distributions.

Thank you for this suggestion. We believe that including information from the null distributions would add a lot of potentially confusing details to this figure. In Fig. S3, we show distributions for the metrics for all populations/years, under the three null models. We now include text in the caption of Fig. 2 that alters the reader to this figure.

“See Fig. S3 for distributions of these metrics under three null models (further described in the methods section).”

Figure 3: What is the null distribution of repeatability if individual’s yearly specialization was sampled from alternative individual’s resource selection functions (but constraining home range size?)

Thank you for this suggestion. We calculated repeatability under all three of the null models and compared their distributions to the observed value. Please see R9 and Fig. S7 for additional details.

In the supplement we learn that there was an “interaction term for percent forest and NDVI”. Why is this not in the results, and why was this included?

We now discuss the interaction term in several places in the text. Please see R2 for more details.

Supplemental figures appear to be multiple figures for every caption, making it very difficult to identify which figures are “S1” and “S2”.

We have named the files e.g. “S1” and “S2”. These file names correspond to the caption with the same label in the supplementary document.

S3. P can not equal 0.

We calculated bootstrap p-values using e.g. (# cases < observed value)/total # of cases, where the total number of cases is 100 (Gotelli & Ellison, 2013). When none of the 100 cases are < the observed value, we now report this as $p < 0.01$.

S4. Provide best fit line, and significance of the relationship. “diver” -> “driver”.

Our intention for this figure was to show there was a weak relationship between home range size and niche volume. We believe adding a best fit line to the figure, given such a weak relationship, would be confusing to the readers.

Thank you for pointing out the spelling error we have updated the figure caption.

Table S1: My supposition is that this is actually supposed to be “Table 1” referred to in text? RE: Bare ground. My understanding of the Hansen 2013 dataset was that it included tree cover, and non-tree cover, not bare ground per se. Is this not correct?

Thank you for pointing out this error, you are correct that Table 1 should be Table S1. We have updated this reference.

The percent tree and bare ground data are available at the University of Maryland's Global Land Analysis & Discovery site.

Percent tree cover: <https://glad.umd.edu/dataset/global-2010-tree-cover-30-m>

Percent bare ground: <https://glad.umd.edu/dataset/global-2010-bare-ground-30-m>

We have updated the table with these links.

The suggested citation for these layers is (Hansen et al., 2013). The reviewer is correct that the primary layers from this project is yearly binary forest loss layers, however it seems these percent tree and bare ground layers were produced as part of this effort.

In the maintext the rationale behind these 'niche axes' could be better described. Is "Distance to X" an actual niche axis in the sense of Hutchinson etc? These variables feel like widely available globally sensed data that presumably correlate with the actual niche axes that this species is responding to, but that do not actually in and of themselves represent true niche axes. Justification for use of distance as a niche axis in particular needs to be justified.

We have provided discussion related to this comment above. Please see R5 for additional information.

Response to last round of review:

As you state in your response to reviewers applying the hutchinsonian niche concept to individuals runs into problems in so far as individuals do not have population growth rates. However, the closest thing to population level growth at the individual is fitness.

You are presumably estimating the relationship of fitness proxies (with the associated assumptions therein). If your "individual niches" do not track individual absolute fitness then they are fairly useless as niche measurements.

"Thus, we do not simply consider any environmental conditions the individual experiences as part of its niche, but only those conditions that allow the individual to survive and reproduce. This is one reason why we focus on foraging locations during the breeding period in our study, and have filtered out other locations such as sitting, preening, flying, etc. By focusing on the environmental characteristics of an individual's foraging locations, which are closely tied to the individual's ability to survive and reproduce, we estimate individual realized niches."

These statements are fine but they are prone to miss key components of the environment that will affect survival and reproduction other than just foraging.

We agree that processes other than foraging affect the ability of an individual to survive and reproduce. For example, a leading cause of death in storks is due to electric wires, and eggs are often broken during fights at the nest with conspecifics. However, these events are rare relative

to the amount of time storks spend foraging and feeding their chicks. So, while there are certainly many other important factors that affect survival and reproduction, by focusing on foraging habitats we capture arguably the most important process that provides information about the characteristics of individual Grinnellian niches.

“Based on these definitions, it is easy to see that our study, which examines environmental axes (such as tree cover, or distance to urban) that are not changed by the presence of individuals (at the time scale of our study) estimate individual Grinnellian niches.” More nuance in paper is needed to acknowledge that the Grinnellian niche variables measured are correlative, and any apparent Grinnellian niche shifts could in fact be due to other environmental variables, or to the Eltonian niche variables as originally suggested by reviewer two. If these variables represent niche axes for storks I certainly agree that they belong in the Grinnellian category. However, not every scenopoetic variable that correlates with presence/survival/reproduction is a relevant niche axis. To give a stupid made up example, the local abundance of Samsung brand cell phones is (presumably) not a real niche axis for these storks (5G conspiracies notwithstanding), even though stork occurrence, abundance, survival, and reproduction almost certainly all correlate with this variable. This undermines your claims that your land-cover variables are definitively relevant niche axes for these storks, as (like with cell phones) the apparent specialization on them (or away from them) could be due their correlation with other true niche variables.

We have softened our language throughout the paper regarding niche axes. Based on knowledge of stork biology, we strongly believe these are niche axes, and resource selection analysis provides no evidence that these are *not* niche axes. But the reviewer is correct, we can't definitively say these are niche axes without performing manipulative experiments examining the relationship between the variables and individual survival and reproduction. However, we believe that based on the significant literature on stork foraging and habitat selection, our own experience studying storks, and bolstered by statistical rejection of the null hypothesis in our resource selection analysis, we believe it is reasonable to assume that these variables are, or are closely correlated with, true niche axes, such that even if we did have the true niche axes (and they were in fact different from the axes we use in this study), our results would be similar.

The Reviewer's example of Samsung phone density makes an important point regarding why it is inadvisable to simply try every available environmental variable and choose those that are correlated with the response. We believe selection of niche axes has two important components. First, an investigator must carefully consider the question and the biology of the target organism, using expert knowledge to select appropriate niche axes. Second, some statistical analysis should be applied to reject niche axes that were inappropriately chosen. In the Reviewer's example, a variable measuring the density of Samsung phones would never be selected in step one, thus it would never be considered a potential niche axes for step two.

Reviewer #4 (Remarks to the Author):

I have read with great interest the manuscript titled “Individual environmental niches in 1 mobile organisms” by Carlson et al. et al.

I did notice this was a revised version, and I had access to all previous review material, including Reviewers’ comments, Authors’ rebuttal, and revised manuscript.

We thank the reviewer for their review and helpful feedback.

I have some more comments that should be addressed to cement the clarity of the paper:

L63-64 this is confusing because you do not in fact address the connection between Eltonian (trophic) and Grinnellian (geophysical) niches later on, but model individual Grinnellian niches and discuss their relative position within the population-level Grinnellian niche.

Thank you for this comment. We have updated the paragraph to include a better topic sentence that more clearly describes the purpose of paragraph. The goal of the paragraph is to describe the importance of direct assessment of Grinnellian niches. We provide background information on the connection between Eltonian and Grinnellian niches in order to show the reader that the relationship is potentially complicated, so we can’t simply assess Eltonian niches and assume Grinnellian niches have the same pattern. We have updated the beginning of the paragraph as follows:

“It is important to directly assess Grinnellian niches and not simply assume that individual niche patterns follow those of Eltonian niches. Although we expect that an individual’s Grinnellian and Eltonian niches are interconnected, this relationship is complicated through a complex set of factors involving the individual’s (and, in secondary consumers, putative prey’s) behavioral response to the environment, and the scale at which environmental associations are assessed.”

L389 The Discussion ends abruptly. I would add a take-home message paragraph based on your data and results, rather than ending on recommendations for future research.

Thank you for this suggestion. While this sort of concluding paragraph is certainly a valid strategy, we worry that the discussion is already a bit long and that adding an additional paragraph would only make it longer. One of our goals with this study is to highlight our approach as a new way to assess individual specialization within populations using remote sensing and tracking data, and thus without requiring intensive field sampling, such as examination of stomach contents. We think that advances in tracking and remote sensing will only make our approach increasingly feasible and useful, and wanted to end on this exciting note. We hope the reviewer is satisfied with this approach.

L429 I don’t understand why you even need to define a home range if you perform a step-selection function? The domain of availability is defined by the step-length distribution, and there is no need to build a home range; but I might have missed something.

The reviewer is correct to point out that the available distribution for the SSF is defined by sampling environments around each observed location using the step-length distribution. Thus, our home range was not used in SSF analysis, but instead were intended to illustrate to readers the spatial distribution of individuals. We believe these home ranges visually demonstrate that individuals within each population had a high degree of actual or potential overlap in space use. Coupled with knowledge of these animal's movement capacity, we believe the ranges provide visual evidence that storks within a population had access to similar environments.

L 443 SSFs are not themselves niche models, but...

We changed "SSFs are not themselves niches, but..." to "SSFs do not directly model niches, but ..." We hope this is in keeping with the reviewer's helpful suggestion.

L444 do you have a reference to back up this approach of using a SSF to identify niche axes?

To our knowledge, we are the first to make the small conceptual advance from using SSFs to infer habitat selection to interpreting selection as evidence for niche axes that are used in the construction of hypervolumes. We believe this is a reasonable interpretation of SSF analysis and is in keeping with the significant body of existing SSF literature.

The methods in the Environmental conditions and Resource selection sections are not very clear. Maybe swapping these 2 sections around could help. First you present the RS analysis, and then you introduce the environmental variables (the term conditions is a bit odd in this context) at what is now L495, between the amt packages and distance to nest.

In this study, we felt that it is important to stress our niche axes and their use in the construction of hypervolumes. While the resource selection analysis was an important methodological step in supporting our *a priori* choice of environmental variables, we feel that describing resource selection first would unduly highlight the importance of this analysis over the hypervolume analysis.

In response to other feedback, we updated these paragraphs. We hope these updates clear up any confusion that prompted this comment. We are happy to edit the contents of these paragraphs further if the reviewer could provide more specific details about what was unclear.

485 ... we performed a resource selection analysis. Resource selection analyses seek to ...

We refer to "resource selection analysis" as an analysis category that includes a broad suite of techniques (including resource selection functions and step selection functions). Thus, we don't believe that adding the indefinite article "a" is necessary from a grammatical standpoint or feel it adds any clarification to the sentence.

L505 intervals did not overlap 0

We changed “confidence intervals did not contain 0” to the suggested “intervals did not overlap 0”

L517 we used to calculate

Thank you for pointing out this error. We updated “used to calculated” to “used to calculate”

- Blonder, B., Morrow, C. B., Maitner, B., Harris, D. J., Lamanna, C., Violle, C., Enquist, B. J., & Kerkhoff, A. J. (2018). New approaches for delineating n-dimensional hypervolumes. *Methods in Ecology and Evolution*, *9*(2), 305–319. <https://doi.org/10.1111/2041-210X.12865>
- Bolnick, D. I., Svanbäck, R., Fordyce, J. A., Yang, L. H., Davis, J. M., Hulse, C. D., Forister, M. L., & McPeck, A. E. M. A. (2003). The Ecology of Individuals: Incidence and Implications of Individual Specialization. *The American Naturalist*, *161*(1), 1–28. <https://doi.org/10.1086/343878>
- Carpenter, J., Aldridge, C., & Boyce, M. S. (2010). Sage-Grouse Habitat Selection During Winter in Alberta. *The Journal of Wildlife Management*, *74*(8), 1806–1814. <https://doi.org/10.2193/2009-368>
- Carrascal, L. M., Alonso, J. C., & Alonso, J. A. (1990). Aggregation size and foraging behaviour of white storks *Ciconia ciconia* during the breeding season. *Ardea*, *78*, 399–404.
- Chetkiewicz, C.-L. B., & Boyce, M. S. (2009). Use of resource selection functions to identify conservation corridors. *Journal of Applied Ecology*, *46*(5), 1036–1047. <https://doi.org/10.1111/j.1365-2664.2009.01686.x>
- Clark, J. D., Laufenberg, J. S., Davidson, M., & Murrow, J. L. (2015). Connectivity among subpopulations of louisiana black bears as estimated by a step selection function. *The Journal of Wildlife Management*, *79*(8), 1347–1360. <https://doi.org/10.1002/jwmg.955>
- Duduś, L., Zalewski, A., Koziół, O., Jakubiec, Z., & Król, N. (2014). Habitat selection by two predators in an urban area: The stone marten and red fox in Wrocław (SW Poland). *Mammalian Biology*, *79*(1), 71–76. <https://doi.org/10.1016/j.mambio.2013.08.001>
- Gotelli, N. J., & Ellison, A. M. (2013). *A Primer of Ecological Statistics*. Sinauer.
- Hansen, M. C., Potapov, P. V., Moore, R., Hancher, M., Turubanova, S. A., Tyukavina, A., Thau, D., Stehman, S. V., Goetz, S. J., Loveland, T. R., Kommareddy, A., Egorov, A., Chini, L., Justice, C. O., & Townshend, J. R. G. (2013). High-Resolution Global Maps of 21st-Century Forest Cover Change. *Science*, *342*(6160), 850–853. <https://doi.org/10.1126/science.1244693>
- Hertel, A. G., Leclerc, M., Warren, D., Pelletier, F., Zedrosser, A., & Mueller, T. (2019). Don't poke the bear: Using tracking data to quantify behavioural syndromes in elusive wildlife. *Animal Behaviour*, *147*, 91–104. <https://doi.org/10.1016/j.anbehav.2018.11.008>

- Hutchinson, G. E. (1978). *An introduction to population ecology*.
<http://www.ponline.org/node/442936>
- Johnson, D. S., Thomas, D. L., Hoef, J. M. V., & Christ, A. (2008). A General Framework for the Analysis of Animal Resource Selection from Telemetry Data. *Biometrics*, *64*(3), 968–976.
<https://doi.org/10.1111/j.1541-0420.2007.00943.x>
- Marzlufi, J. M., & Heinrich, B. (1991). Foraging by common ravens in the presence and absence of territory holders: An experimental analysis of social foraging. *Animal Behaviour*, *42*(5), 755–770. [https://doi.org/10.1016/S0003-3472\(05\)80121-6](https://doi.org/10.1016/S0003-3472(05)80121-6)
- Moore, S. A., & Bronte, C. R. (2001). Delineation of Sympatric Morphotypes of Lake Trout in Lake Superior. *Transactions of the American Fisheries Society*, *130*(6), 1233–1240.
[https://doi.org/10.1577/1548-8659\(2001\)130<1233:DOSMOL>2.0.CO;2](https://doi.org/10.1577/1548-8659(2001)130<1233:DOSMOL>2.0.CO;2)
- Nakagawa, S., & Schielzeth, H. (2010). Repeatability for Gaussian and non-Gaussian data: A practical guide for biologists. *Biological Reviews*, *85*(4), 935–956.
<https://doi.org/10.1111/j.1469-185X.2010.00141.x>
- Nielson, R. M., & Sawyer, H. (2013). Estimating resource selection with count data. *Ecology and Evolution*, *3*(7), 2233–2240. <https://doi.org/10.1002/ece3.617>
- Piper, W. H. (1997). Social Dominance in Birds. In V. Nolan, E. D. Ketterson, & C. F. Thompson (Eds.), *Current Ornithology* (pp. 125–187). Springer US. https://doi.org/10.1007/978-1-4757-9915-6_4
- Rainho, A., & Palmeirim, J. M. (2011). The Importance of Distance to Resources in the Spatial Modelling of Bat Foraging Habitat. *PLOS ONE*, *6*(4), e19227.
<https://doi.org/10.1371/journal.pone.0019227>
- Schielzeth, H. (2010). Simple means to improve the interpretability of regression coefficients. *Methods in Ecology and Evolution*, *1*(2), 103–113. <https://doi.org/10.1111/j.2041-210X.2010.00012.x>
- Thurfjell, H., Ciuti, S., & Boyce, M. S. (2014). Applications of step-selection functions in ecology and conservation. *Movement Ecology*, *2*(4).
<http://www.biomedcentral.com/content/pdf/2051-3933-2-4.pdf>
- van Overveld, T., García-Alfonso, M., Dingemanse, N. J., Bouten, W., Gangoso, L., de la Riva, M., Serrano, D., & Donazar, J. A. (2018). Food predictability and social status drive individual resource specializations in a territorial vulture. *Scientific Reports*, *8*(1), 15155.
<https://doi.org/10.1038/s41598-018-33564-y>

REVIEWERS' COMMENTS

Reviewer #3 (Remarks to the Author):

The authors have addressed my major concerns. Their edits have produced a stronger and more clear manuscript that will influence thinking on inter-individual niche variation.

Reviewer #4 (Remarks to the Author):

I have read the revised manuscript and the replies to reviewers' comment - I think the manuscript is now clearer; I congratulate the Authors for their effort in revising their manuscript.